# Metabolic dysregulation contributes to the development of dysferlinopathy

Regula Furrer[1], Sedat Dilbaz[1], Stefan A Steurer[1], Gesa Santos[1], Bettina Karrer-Cardel[1], Danilo Ritz[1], Michael Sinnreich[2], Christoph Handschin[1]

**Dysferlin is a transmembrane protein that plays a prominent role in membrane repair of damaged muscle fibers. Accordingly, mutations in the dysferlin gene cause progressive muscular dystrophies, collectively referred to as dysferlinopathies for which no effective treatment exists. Unexpectedly, experimental approaches that successfully restore membrane repair fail to prevent a dystrophic phenotype, suggesting that additional, hitherto unknown dysferlin-dependent functions contribute to the development of the pathology. Our experiments revealed an altered metabolic phenotype in dysferlin-deficient muscles, characterized by (1) mitochondrial abnormalities and elevated death signaling and (2) increased glucose uptake, reduced glycolytic protein levels, and pronounced glycogen accumulation. Strikingly, elevating mitochondrial volume density and muscle glycogen accelerates disease progression; whereas, improvement of mitochondrial function and recruitment of muscle glycogen with exercise ameliorated functional parameters in a mouse model of dysferlinopathy. Collectively, our results not only shed light on a metabolic function of dysferlin but also imply new therapeutic avenues aimed at promoting mitochondrial function and normalizing muscle glycogen to ameliorate dysferlinopathies, complementing efforts that target membrane repair.**

## Introduction

Dysferlinopathies refer to a group of rare, recessively inherited muscular dystrophies that are caused by mutations in the dysferlin gene (*DYSF*), leading to a deficiency or dysfunction of the dysferlin protein (Bashir et al, 1998; Mercuri et al, 2019). Dysferlin is a 237 kD transmembrane protein and plays an essential role in maintaining the structural integrity of the muscle membrane (Ivanova et al, 2022). More specifically, the main function attributed to dysferlin is the involvement in membrane repair (Bansal et al, 2003). As skeletal muscles experience high mechanical stress because of repeated contraction-relaxation cycles, different programs contribute to adequate repair and regeneration and thereby ensure retrieval of

proper fiber morphology and function. Micro-lesions in the plasma membrane result in $Ca^{2+}$ entry, triggering the recruitment of vesicles to the damaged area. Therefore, patches are formed, and the membrane resealed by vesicle-vesicle and vesicle-membrane fusion (Han & Campbell, 2007). Dysferlin is critical for vesicle trafficking and membrane fusion in this process (Han & Campbell, 2007). Accordingly, in dysferlin-deficient muscles, membrane resealing is substantially compromised, leading to the accumulation of damaged muscle fibers and degeneration of muscle tissue, eventually resulting in the development of progressive muscular dystrophy (Bansal et al, 2003). Clinical features of dysferlinopathies typically manifest during late adolescence or early adulthood and include high serum creatine kinase (CK) levels and muscle weakness (Bansal & Campbell, 2004). Histologically, dysferlin-deficient muscles resemble those of other muscular dystrophies, characterized by fiber atrophy, centralized nuclei, immune cell infiltration, and replacement of muscle by fat and fibrotic tissue (Bansal et al, 2003). Patients usually lose independent ambulation ~10 yr after disease onset (Cardenas et al, 2016).

Currently, no curative treatment exists for patients with dysferlinopathies. Because full-length dysferlin cDNA exceeds the packaging capacity of an adeno-associated virus (AAV), a truncated construct of the protein, so-called "mini-dysferlin," containing the last two C2 domains and the transmembrane domain at the C-terminus of the protein, was successfully transduced into dysferlin-deficient mice and able to restore membrane repair (Krahn et al, 2010). Surprisingly, the development of the dystrophic muscle in mice was not prevented by this strategy (Krahn et al, 2010; Lostal et al, 2012). This suggests that besides the role in membrane repair, dysferlin is engaged in other cellular processes that contribute to proper muscle function, which, in the pathological setting, could be involved in disease etiology and progression.

In addition to gene replacement, other features of the pathology have been targeted in preclinical and clinical models; for example, the mitigation of the aberrant immune response or muscle atrophy using drug-based therapies (Walter et al, 2013; Dillingham et al, 2015; Lee et al, 2015). However, none of these interventions were sufficient to reduce the dystrophy or improve muscle function, at least when attempted individually. It, however, is unexplored

---

[1]Biozentrum, University of Basel, Basel, Switzerland    [2]Department of Biomedicine and Neurology, University and University Hospital Basel, Basel, Switzerland

Correspondence: christoph.handschin@unibas.ch

whether simultaneous modulation of multiple disease-specific features (i.e., inflammation, impaired regeneration, and fiber atrophy) could improve the pathology.

Peroxisome proliferator-activated receptor γ coactivator 1α (PGC-1α) is a master regulator of endurance training adaptation, acting as a transcriptional coactivator and orchestrating a transcriptional network involved in regulating a broad program of muscle plasticity, including mitochondrial function and energy metabolism, antioxidant defense, and fiber innervation (Lin et al, 2002; Furrer et al, 2023a). Because of the pleiotropic effects, PGC-1α also plays a protective role in various contexts of impaired muscle function (Chan & Arany, 2014). For example, muscles overexpressing PGC-1α exhibit an anti-inflammatory environment and have an improved regenerative capacity after repeated muscle injury, resulting in reduced muscle fibrosis (Dinulovic et al, 2016a, 2016b). Moreover, denervation-induced muscle atrophy is prevented by PGC-1α overexpressing muscles (Sandri et al, 2006). Most relevant, in a mouse model for Duchenne muscular dystrophy, muscle PGC-1α induces a remarkable decrease in CK levels, reduction in muscle damage, and overall improvement of histopathology and performance (Handschin et al, 2007). PGC-1α thus emerges as a potential therapeutic target to treat muscular dystrophies and could be a potential candidate for a multi-pronged therapy in dysferlinopathies.

In the present study, we therefore have assessed the molecular, cellular, morphological, and functional consequences of elevated muscle PGC-1α in a mouse model of dysferlinopathy. Diametrically opposite to our hypothesis, PGC-1α exacerbated disease progression in dysferlin-deficient male mice despite the reduction in muscle fiber damage. These unexpected findings, however, have resulted in the identification of an important unsuspected function of dysferlin. Our data indicate that abnormal muscle cell metabolism, caused by mutations in the dysferlin gene, contributes to the disease pathology. More specifically, we observed a pronounced elevation in glucose uptake in dysferlin-deficient muscles that, in the context of lowered glycolytic protein levels, leads to the accumulation of muscle glycogen. Furthermore, muscles lacking dysferlin exhibit mitochondrial abnormalities that result in elevated death signaling. The PGC-1α-dependent increment in glucose uptake, glycogen storage, and mitochondrial volume density likely contributed to the more severe muscular dystrophy. This new aspect of dysferlin biology, with the implications for the pathology of dysferlinopathies, indicates potential novel therapeutic avenues that could decisively complement existing strategies aimed at restoring membrane repair.

## Results

### PGC-1α overexpression in muscle exacerbates the progression of dysferlinopathy

To assess whether elevated muscle PGC-1α ameliorates disease progression of dysferlin-deficient mice, analogous to previous findings in Duchenne muscular dystrophy and other muscle wasting models, we crossed skeletal muscle-specific PGC-1α transgenic mice (Lin et al, 2002) with a model for dysferlinopathy (Bansal et al, 2003;

Wiktorowicz et al, 2015). PGC-1α expression is considerably higher in transgenic mice compared with WT, but the expression level is slightly lower in dysferlin-deficient mice overexpressing PGC-1α compared with PGC-1α transgenic controls (~5 and ~7-fold higher, respectively, compared with WT mice); whereas, the expression is not different in dysferlin-deficient mice compared with WT (Fig S1A). In these animals, we first measured CK levels between 4 and 15 mo of age. Throughout this time period, CK levels are substantially elevated in the absence of dysferlin, and this elevation was not affected by higher levels of muscle PGC-1α (Fig 1A). The rise in plasma CK is accompanied by a progressive loss of muscle mass. The proximal psoas muscle is most severely affected by the disease, with a significant reduction in mass (~25% lower compared with WT animals after 7, and ~70% at the age of 15 mo) (Fig 1B) and is followed by the muscle mass loss of the gluteus that starts atrophying at the age of 7 mo (Fig S1B). The quadriceps muscle, comparatively to the psoas and gluteus more distal, shows the first signs of atrophy at the age of 11 mo (Fig 1B). In contrast, distal muscles of the lower limb, including the extensor digitorum longus (EDL), soleus and tibialis anterior (TA) were not affected by atrophy, or even depicting compensatory hypertrophy (Figs 1C and S1C). Hence, there is no disease preference towards a certain fiber type (fast- versus slow-twitch muscle fibers) but rather towards location (proximal versus distal). In humans, there are two widely used clinical diagnoses for dysferlinopathy including limb girdle muscular dystrophy (LGMD) 2B/R and Myoshi myopathy (Moore et al, 2021). Whereas the manifestation of these diseases may vary at disease onset by affecting more proximal or distal muscles, respectively, there is a large phenotypic overlap and both diseases have, therefore, been proposed to be referred to as dysferlinopathies (Moore et al, 2021). However, it appears that in mice, there is a clear LGMD2B/R phenotype and even with advanced age, the distal muscles are completely protected from atrophy. Therefore, it is important to consider this aspect when studying mouse muscles.

The presence of elevated muscle PGC-1α produced a mixed outcome: no effect on most muscles and time points, a slight protection of the gluteus at 15 mo, an exacerbation of the compensatory hypertrophy in the soleus and TA muscles at 11 and 15 mo, and an acceleration of quadriceps atrophy at 11 mo of age (Figs 1B and C and S1B and C). Thus, opposite to our hypothesis, muscle PGC-1α does not broadly ameliorate disease progression in terms of muscle mass loss. Strikingly, an aggravation of the pathology by PGC-1α was seen in some functional parameters such as grip strength, running wheel locomotor activity and gait; whereas, others were unaffected (Fig S1D and E). Collectively, these results indicate the absence of a broad therapeutic effect of elevated muscle PGC-1α on dysferlinopathy and in some cases, even an acceleration of disease progression.

Because the quadriceps muscle manifests a pronounced loss of muscle mass at the age of 11 mo and exhibits a surprising difference between the genotypes, we performed in-depth analysis of this muscle. First, we assessed muscle damage in the quadriceps muscle by Evans blue dye incorporation. Strikingly, opposite to the exacerbated loss of muscle mass, dysferlin-deficient mice overexpressing PGC-1α exhibit less Evans blue-positive fibers at the age of 11 mo (Fig 1D and E). This suggests that despite the accelerated atrophy, muscle PGC-1α mitigates muscle fiber membrane damage in the absence of dysferlin. However, in stark contrast, histopathological analyses

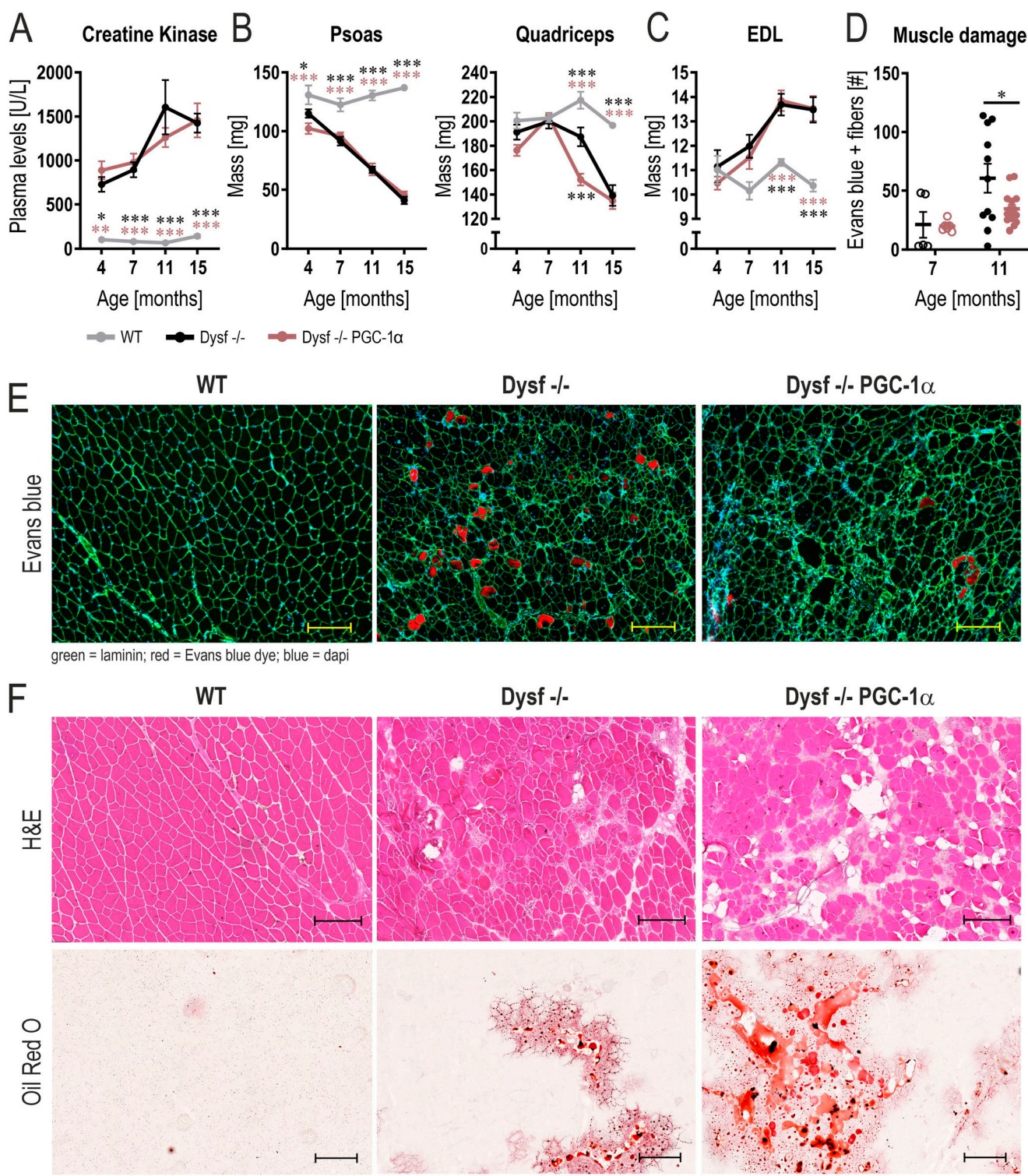

**Figure 1. PGC-1α overexpression in muscle exacerbates the progression of dysferlinopathy.**
**(A)** Creatine kinase levels in plasma at different ages in WT mice, mice lacking dysferlin (Dysf −/−) and Dysf −/− mice overexpressing PGC-1α in muscle (Dysf −/− PGC-1α).
**(B, C)** Trajectories of the mass of proximal muscles (B) and the distal muscle extensor digitorum longus (EDL) in panel (C). **(D)** Number of Evans blue-positive fibers in the rectus femoris of 11 mo old mice. **(E)** Representative examples of histological sections of rectus femoris muscle of 11 mo old mice that were injected with Evans blue dye.
**(F)** Representative examples of H&E or Oil Red O stained sections of rectus femoris muscle of 11 mo old mice. Data information: data represent means ± SEM; scale bars = 200 μm. For panel (A), data are included from WT 4 mo n = 6, WT 7 mo n = 12, WT 11 mo n = 11, WT 15 mo n = 7, Dysf −/− 4 mo n = 7, Dysf −/− 7 mo n = 12, Dysf −/− 11 mo n = 12, Dysf −/− 15 mo n = 8, Dysf −/− PGC-1α 4 mo n = 6, Dysf −/− PGC-1α 7 mo n = 13, Dysf −/− PGC-1α 11 mo n = 16, Dysf −/− PGC-1α 15 mo n = 8. **(B)** For the psoas muscle in panel (B), data are included from WT 4 mo n = 7, WT 7 mo n = 5, WT 11 mo n = 6, WT 15 mo n = 8, Dysf −/− 4 mo n = 7, Dysf −/− 7 mo n = 7, Dysf −/− 11 mo n = 10, Dysf −/− 15 mo n = 8, Dysf −/−

using H&E and Oil Red O staining reveal a more severe dystrophic phenotype (Fig 1F). Thus, even though our data suggest that elevated muscle PGC-1α protects myofiber membrane integrity, an exacerbated disease phenotype is observed in the quadriceps muscles of the PGC-1α overexpressing dysferlin-deficient mice, with accelerated muscle mass loss and adipogenic replacement of muscle tissue.

## Dysregulation of metabolism in dysferlin-deficient muscles

The exacerbation of pathological parameters by PGC-1α could imply a conversion of molecular mechanisms regulated by PGC-1α and ablated dysferlin independent of membrane repair, that, when combined, lead to a worse outcome. Thus, to obtain insights into such processes that drive accelerated disease progression, we performed a comparative proteomic analysis of dysferlin-deficient quadriceps muscle (11 mo) and age-matched, healthy muscles overexpressing PGC-1α (Furrer et al, 2023b). The proteome in either genotype is substantially different to that of healthy WT muscles (Fig S2A and B; Table S1). Surprisingly, despite the very different phenotype of these two mouse models, ~25% of all up-regulated and 56% of all down-regulated proteins in the dysferlin-deficient muscles overlap with the changes seen in healthy muscles of muscle-specific PGC-1α transgenic mice (Fig 2A). A significant part of the up-regulated proteins found in both models is involved in mitochondrial translation, exhibiting a pronounced increase in mitochondrial ribosomal proteins (Figs 2B and S3A). On the other hand, the proteins with a lower abundance contribute to muscle contraction, glycolytic process, and glycogen metabolic process (Fig 2B). Taken together, the changes in the proteome indicate a broad shift from glucose metabolism to mitochondrial respiration, a prominent feature of muscle metabolism in the PGC-1α transgenic animals (Lin et al, 2002; Wende et al, 2007), but unexpected in the dysferlinopathy mice.

To further elucidate this potential metabolic shift, we assessed mitochondrial morphology and function. Electron microscopy analyses revealed that, whereas mitochondrial volume density is unchanged in muscles lacking dysferlin, mitochondrial number and size are altered (Fig 2C and D). Dysferlin-deficient muscles contain fewer mitochondria; however, these mitochondria are on average larger in size (Fig 2D). Accordingly, the proportion of relatively large mitochondria is also higher (Fig 2E), suggesting that mitochondrial dynamics might be affected in these muscles, even though the abundance of most proteins involved in fission and fusion is unchanged in the absence of dysferlin (Fig S3B). We next studied whether these morphological differences also result in distinct expression of proteins involved in mitochondrial respiration. Despite the same mitochondrial volume density, proteins involved in the tricarboxylic acid (TCA) cycle and of the electron transport chain are lower in muscles lacking dysferlin (Figs 2F, S2B, and S3C).

Whereas there was no significant reduction in mitochondrial citrate synthase activity and mitochondrial respiration, there was a trend towards a reduced function (Fig S3D and E). Collectively, these data imply that the absence of dysferlin promotes mitochondrial abnormalities.

Because muscles overexpressing PGC-1α have a higher mitochondrial density and mitochondrial function is altered in muscles lacking dysferlin, the question arises whether this mitochondrial phenotype could drive the pathology in dysferlin-deficient muscles overexpressing PGC-1α. Morphological analysis of electron microscopy images revealed that mitochondrial volume density, number of mitochondria and size are unaffected in the absence of dysferlin (Fig S4A–C). However, the abundance of proteins involved in mitochondrial respiration is largely reduced, suggesting that the function of these mitochondria might be impaired (Fig S4D and E). Indeed, both citrate synthase activity and mitochondrial respiration (complex II and IV) are substantially lower in dysferlin-deficient mice overexpressing PGC-1α compared with PGC-1α controls, reaching levels close to those seen in WT animals (Fig S4F and G). This is surprising considering that, in this context, PGC-1α expression is five times higher compared with WT animals (Fig S1A). These results imply an epistatic relationship between PGC-1α and dysferlin, at least in the control of mitochondrial protein expression. The unaltered mitochondrial volume density and concomitant decrease in mitochondrial proteins and function implies a disconnect between morphology and function, suggesting dysfunctional mitochondria that could influence cellular health.

Disruption of $Ca^{2+}$ homeostasis can affect mitochondrial activity (Gherardi et al, 2021). In the absence of dysferlin, membrane repair is defective, with a concomitant prolonged elevation of intracellular $Ca^{2+}$ post-injury (Defour et al, 2014). Furthermore, both dysferlin and PGC-1α play a role in $Ca^{2+}$ handling (Summermatter et al, 2012; Quinn et al, 2024). Therefore, we assessed $Ca^{2+}$ concentration and the abundance of proteins involved in $Ca^{2+}$ handling. Total $Ca^{2+}$ concentration in muscle is elevated in muscles overexpressing PGC-1α, but is not affected by the absence of dysferlin (Fig S5A). In healthy muscles overexpressing PGC-1α, sarco/endoplasmic reticulum $Ca^{2+}$-ATPase (SERCA) activity is reduced and accompanied by a slower re-uptake of cytosolic $Ca^{2+}$ (Summermatter et al, 2012). Interestingly, the abundance of proteins involved in $Ca^{2+}$ handling changes in the same direction in dysferlin-deficient muscles and those overexpressing PGC-1α (Fig S5B). For example, levels of ryanodine receptor 1 (RYR1), SERCA1, calsequestrin 1 (CASQ1), and parvalbumin (PVALB) are lower; whereas, those of CASQ2 and mitochondrial $Ca^{2+}$ uniporter (MCU) are higher, implying similar $Ca^{2+}$ handling properties in these mouse lines (Figs 2G and S5B). The change in the levels of many of these proteins is in line with a preference for oxidative muscle fibers (Summermatter et al, 2012). In line, the most down-regulated common term "muscle contraction"

---

PGC-1α 4 mo n = 6, Dysf −/− PGC-1α 7 mo n = 7, Dysf −/− PGC-1α 11 mo n = 9, Dysf −/− PGC-1α 15 mo n = 8. For the quadriceps muscle in panel (B), data are included from WT 4 mo n = 7, WT 7 mo n = 13, WT 11 mo n = 14, WT 15 mo n = 8, Dysf −/− 4 mo n = 7, Dysf −/− 7 mo n = 12, Dysf −/− 11 mo n = 16, Dysf −/− 15 mo n = 8, Dysf −/− PGC-1α 4 mo n = 6, Dysf −/− PGC-1α 7 mo n = 13, Dysf −/− PGC-1α 11 mo n = 16, Dysf −/− PGC-1α 15 mo n = 8. For panel (C), data are included from WT 4 mo n = 7, WT 7 mo n = 8, WT 11 mo n = 14, WT 15 mo n = 8, Dysf −/− 4 mo n = 7, Dysf −/− 7 mo n = 5, Dysf −/− 11 mo n = 16, Dysf −/− 15 mo n = 8, Dysf −/− PGC-1α 4 mo n = 6, Dysf −/− PGC-1α 7 mo n = 6, Dysf −/− PGC-1α 11 mo n = 16, Dysf −/− PGC-1α 15 mo n = 8. **(A, B, C, D)** Asterisks indicate differences between the genotypes of the same age using two-way ANOVA with Šídák's multiple comparisons test when comparing two groups and Tukey's multiple comparisons test when comparing three groups (the different color of the asterisks in panels (A, B, C) indicate the comparison group); *$P$ < 0.05; **$P$ < 0.01; ***$P$ < 0.001.

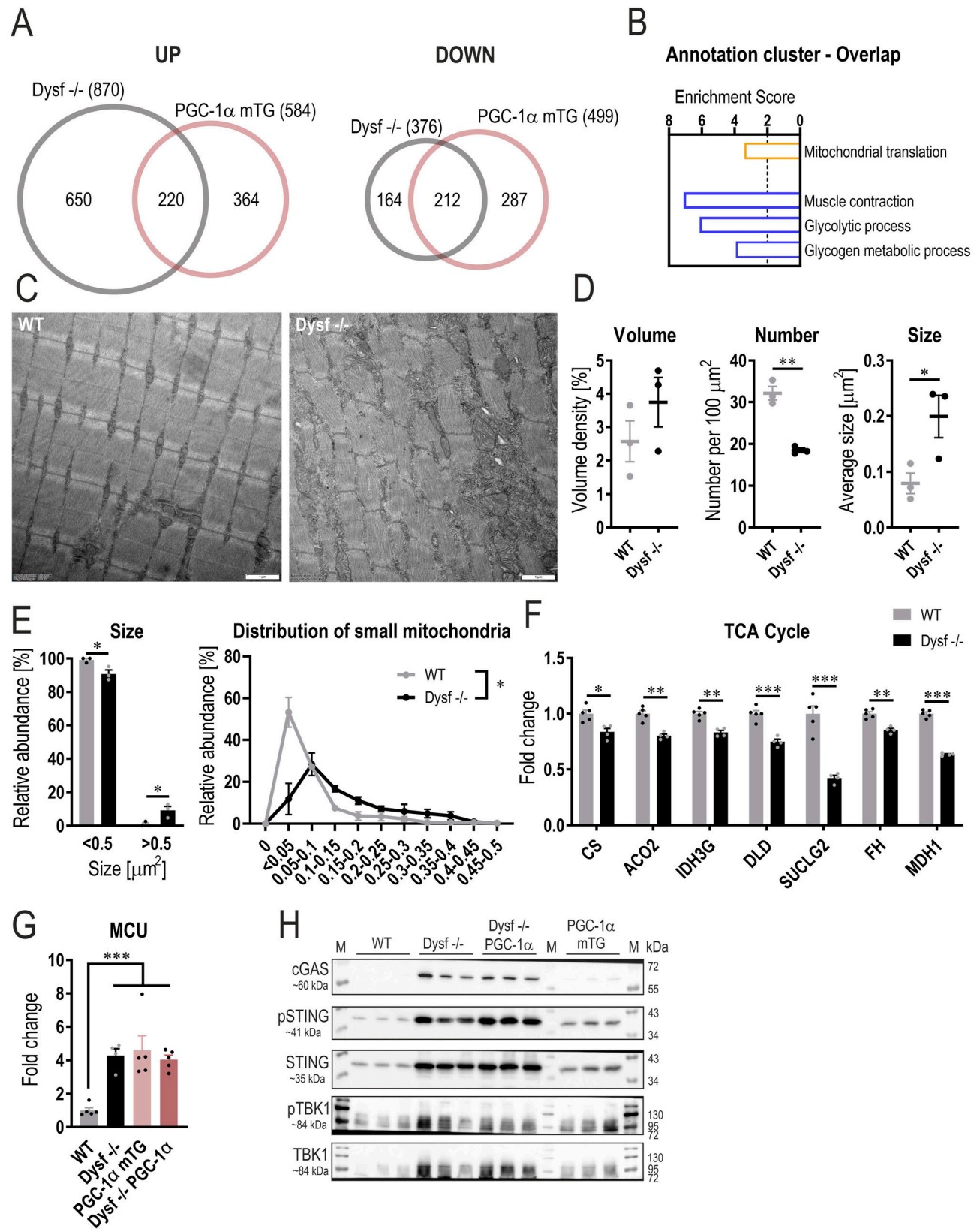

**Figure 2. Mitochondrial abnormalities and elevated death signaling in dysferlin-deficient muscles.**
**(A)** Venn diagram of all proteins that are significantly higher or lower abundant in quadriceps muscle of 11 mo old Dysf −/− or PGC-1α mTG mice compared with WT controls (cutoffs: peptide >1; *q*-value < 0.01; Log₂FC ±0.2). **(B)** Functional annotation clusters of all commonly up-regulated (orange) or down-regulated (blue) proteins

(Fig 2B) predominantly contains proteins associated with fast-twitch fibers that are lower abundant in the absence of dysferlin (Fig S5C). For example, myosin heavy chain 4 (MYH4, present in IIB fibers) levels are reduced, diametrically opposite to those of MYH1 (present in IIX fibers) (Fig S5D). Taken together, lack of dysferlin induces a shift in $Ca^{2+}$ handling similar to that observed with PGC-1α overexpression, except that the total $Ca^{2+}$ concentration is only elevated in muscles overexpressing PGC-1α. Based on previous observations made on muscles overexpressing PGC-1α (Summermatter et al, 2012), it is thus conceivable that intracellular $Ca^{2+}$ might be elevated in dysferlin-deficient muscles overexpressing PGC-1α and could contribute to the exacerbation of the pathology.

Interestingly, MCU was elevated in dysferlin-deficient muscles and dysferlin-deficient muscles overexpressing PGC-1α, suggesting that $Ca^{2+}$ uptake into the mitochondria is elevated. This could lead to $Ca^{2+}$ overload in the mitochondria, which can provoke the opening of the mitochondrial permeability transition pore (Gherardi et al, 2021). The subsequent swelling and rupture of the mitochondria can substantially affect cellular health by triggering death signaling. Hence, the increase in dysfunctional mitochondria (in dysferlin-deficient mice overexpressing PGC-1α) might be detrimental for cellular health and promote disease development. To test this hypothesis, we measured the cyclic GMP-AMP (cGAMP) synthase (cGAS)—stimulator of interferon genes (STING) pathway, including the downstream target TANK-binding kinase 1 (TBK1). In muscles lacking dysferlin (both Dysf −/− and Dysf −/− PGC-1α), this pathway is massively enhanced (Figs 2H and S5E), suggesting that this death signaling pathway could substantially contribute to the pathology. Taken together, our data indicate that in the absence of dysferlin, $Ca^{2+}$ overload promotes mitochondrial abnormalities and stimulates the corresponding death signaling pathway.

### Altered glucose metabolism and glycogen accumulation in dysferlin-deficient muscle occurs before muscle atrophy

In addition to mitochondrial abnormalities, processes involved in glycolysis and glycogen metabolism are down-regulated in muscles lacking dysferlin (Fig 2B). Therefore, the abundance of different key proteins involved in glucose metabolism was assessed. First, the levels of the glucose transporter, type 4 (GLUT4), are higher in the absence of dysferlin (Fig 3A). In line, glucose uptake is augmented in dysferlin-deficient quadriceps muscles 45 min after glucose injection (Fig 3B). Of note, increased glucose uptake was seen in the quadriceps, but not in the distal TA muscle that exhibits no atrophy in this dysferlinopathy model (Fig S6A). In accordance with the higher glucose uptake in dysferlin-deficient muscles, blood glucose

levels are cleared more rapidly during a glucose tolerance test (GTT) compared with WT mice (Fig 3C). The faster glucose clearance post-GTT is already observed at the age of 7 mo (Fig S6B). Importantly, increased glucose uptake was not linked to differences in circulating insulin levels (Fig S6C), indicating a major contribution of muscle-intrinsic changes to this process. Next, we assessed the potential fate of glucose in the muscle. As a major contributor to the functional annotation cluster (Fig 2B), protein levels of every enzyme involved in glycolysis are lower in mice lacking dysferlin compared with WT mice (Fig 3D). The concomitant elevation of glucose-6-phosphate dehydrogenase (G6PD) levels (Fig 3E) indicates that at least some of the glucose is redirected from glycolysis to the pentose phosphate pathway. Because biosynthetic and degrading factors for glycogen, e.g., glycogen synthase kinase 3 β (GSK3β), glycogen synthase (GYS1), and phosphorylase (PYGM), are all reduced in dysferlin-deficient muscles (Fig 3F), the net effect on glycogen homeostasis is difficult to predict. Determination of phosphorylation of these proteins revealed a likewise mixed picture (Table S2). When phospho-data are normalized to total protein (Fig S6D), phosphorylation of inhibiting GSK3β site Ser9 is higher in the absence of dysferlin. In line with this result, phosphorylation of the inhibiting site Ser645 of GYS1 is lower, concomitant with a reduced phosphorylation of various sites of PYGM (except the activating site Ser15), suggesting an enhanced synthesis and decreased breakdown of glycogen. However, when not normalizing the data to total protein (Fig S6E), Ser9 phosphorylation of GSK3β is lower; whereas, phosphorylation of Ser645 of GYS1 and various sites of PYGM are still lower. Of note, there are various phosphorylation sites that are altered for which the function is unclear. Collectively, these data nevertheless point towards an elevated synthesis and lower breakdown of glycogen. Taken together, our results demonstrate that in the absence of the dysferlin gene, muscles that are affected by pathological changes exhibit modifications in glucose uptake and usage.

In various aspects, the metabolic phenotype of dysferlin-deficient muscle resembles that seen in healthy mice overexpressing muscle PGC-1α, specifically pertaining to elevated levels of glucose transporters, higher glucose uptake, increased levels of the pentose phosphate pathway enzyme G6PD, and lower levels of glycolytic enzymes and of glycogen phosphorylase (Wende et al, 2007; Summermatter et al, 2010). Because muscle-specific PGC-1α transgenic mice have a substantially higher glycogen content compared with WT animals, we next studied whether the absence of dysferlin would also result in the accumulation of glycogen. Periodic acid-Schiff (PAS) staining revealed that glycogen levels are considerably higher in dysferlin-deficient muscles compared with WT (Fig 3G). This

using DAVID (Database for Annotation, Visualization and Integrated Discovery). **(C)** Representative electron microscopy (EM) images of quadriceps muscle of 11 mo old mice. **(D)** Quantification of the EM pictures assessing mitochondrial volume density, number, and size. **(E)** Distribution of small (<0.5 $\mu m^2$) and large (>0.5 $\mu m^2$) mitochondria and a more detailed distribution of mitochondria <0.5 $\mu m^2$. **(F)** Relative protein abundance (from proteomic data) of enzymes involved in TCA cycle expressed relative to WT levels. **(G)** Relative protein abundance (from proteomic data) of the mitochondrial $Ca^{2+}$ uniporter (MCU). **(H)** Western blot of cyclic GMP-AMP (cGAMP) synthase (cGAS)—stimulator of interferon genes (STING) pathway and the downstream target TANK-binding kinase 1 (TBK1). M represents the protein ladder (marker). The Ponceau S-stained membrane that serves as loading control is presented in Fig S5E. Data information: for the proteomic analysis, n = 4–5 biological replicates. Data represent means ± SEM. **(A, D, E, F, G)** Empirical Bayes moderated t-statistics was used to analyze proteomic data, two-tailed t test for panel (D), and two-way repeated measures ANOVA for panel (E); *$P$ < 0.05; **$P$ < 0.01; ***$P$ < 0.001 (for panel (F, G), asterisks represent q-values). CS, citrate synthase; ACO2, aconitase 2; IDH3G, isocitrate dehydrogenase (NAD) subunit γ; DLD, dihydrolipoamide dehydrogenase; SUCLG2, succinyl-CoA ligase (GDP-forming) subunit β; FH, fumarate hydratase; MDH1, malate dehydrogenase 1.

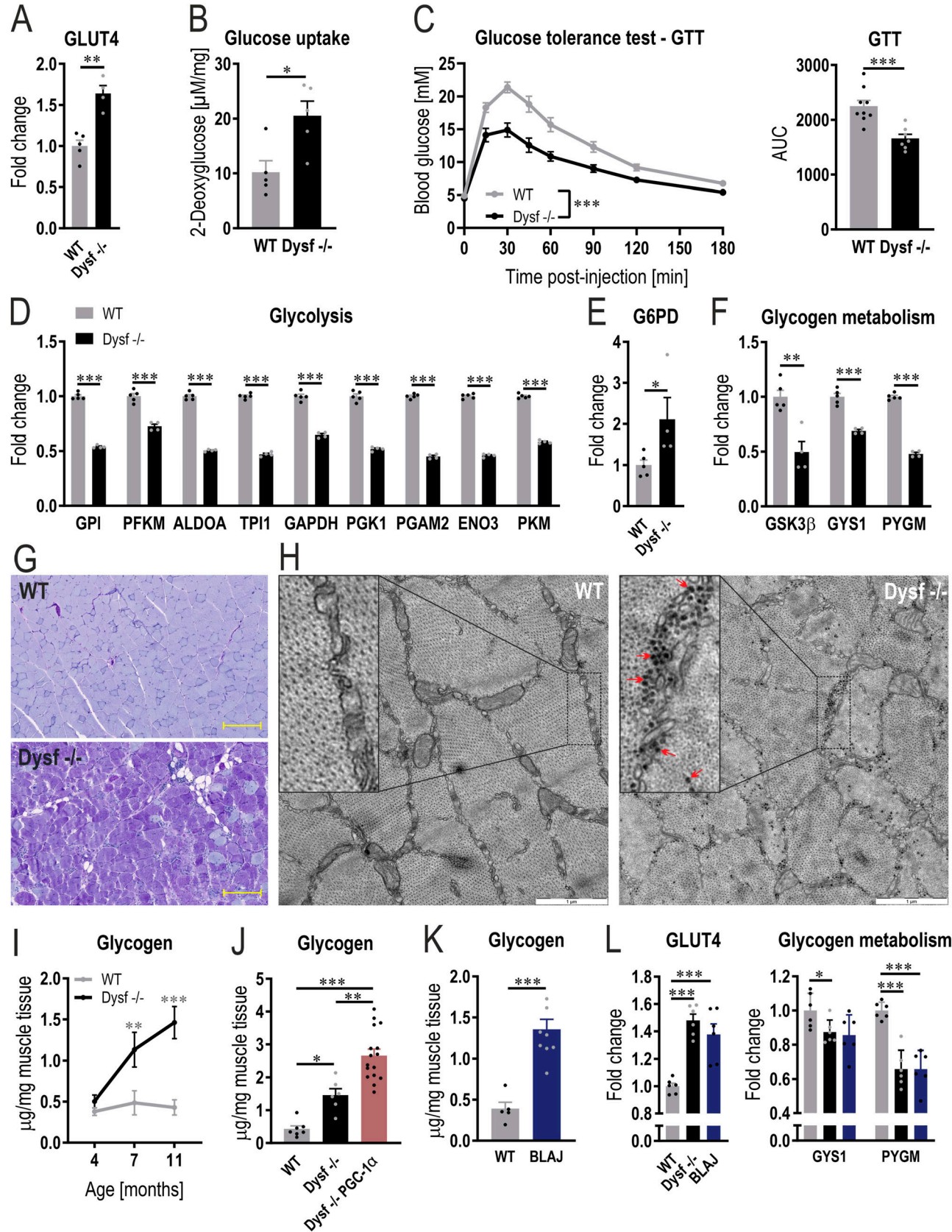

finding could also be confirmed with electron microscopy, displaying more glycogen granules (Fig 3H). Furthermore, we quantified muscle glycogen biochemically and observed that glycogen already accumulates in quadriceps muscle at the age of 7 mo (Fig 3I), thus before muscle atrophy, and in line with the elevated glucose clearance at that age (Figs 1B and S6B). Similar to the normal glucose uptake observed in non-dystrophic muscles, glycogen levels were not elevated in TA muscle (Fig S7A). These results are in line with previous findings demonstrating that the glycogen content of non-dystrophic soleus and EDL muscles is not different to that of WT muscles (Lloyd et al, 2022). In accordance with the role of PGC-1α in promoting glycogen accumulation, dysferlin-deficient mice overexpressing PGC-1α have even higher glycogen content (Fig 3J). Finally, we validated our finding in BLAJ mice (mice carrying the A/J dysferlin mutation that were backcrossed onto the C57BL/6 background), another commonly used mouse model for dysferlinopathies (Grounds et al, 2014; Haynes et al, 2019; Hogarth et al, 2019; Lloyd et al, 2022). At the age of 11 mo, CK levels and muscle mass are very similar in both dysferlin-deficient mouse models (including pronounced atrophy in proximal muscles and compensatory hypertrophy of EDL muscle; Fig S7B–D). Supporting our findings, glycogen levels in quadriceps muscle of 11 mo old BLAJ mice are also elevated (Fig 3K). In accordance with these results, GLUT4 and G6PD protein abundance is higher and levels of glycolytic enzymes as well as PYGM are lower in muscle of BLAJ mice (Figs 3L and S7E and F), similar as seen in Dysf −/− animals (Table S3). This suggests that altered glucose metabolism and concomitant accumulation of glycogen might be directly linked to the absence of dysferlin.

### Exercise ameliorates muscle atrophy and improves performance in dysferlin-deficient mice

A subset of glycogen storage diseases (GSD), such as GSD II (also known as Pompe disease), GSD III (also known as Cori or Forbes disease), or GSD V (also known as McArdle disease), exhibit a myopathic phenotype accompanied by muscle weakness that is caused by high glycogen content in skeletal muscle because of the absence of enzymes involved in glycogen breakdown (i.e., α-1,4-glucosidase [GAA], glycogen debranching enzyme [encoded by the *AGL* gene] or PYGM) (Hicks et al, 2011; van der Beek et al, 2012). In dysferlin-deficient mice, the levels of some of these enzymes are

~40% lower (Fig S7G), which could contribute to reduced glycogen breakdown and, hence, an increase in glycogen storage. Because elevated glycogen levels in the muscle can be detrimental, it is possible that the accumulation of glycogen in muscle lacking dysferlin could contribute to disease progression. Nutritional interventions have been considered to reduce muscle glycogen in GSD. We have tested a glucose-free diet that has been described to substantially reduce muscle glycogen in GSD III mice (Pagliarani et al, 2018). However, in our trial, not only did this diet fail to lower muscle glycogen in mice lacking dysferlin but also exacerbated loss in muscle mass (Table S4). This diet contains more than double the amount of protein compared with the regular chow diet (66 kilojoule [KJ]% compared with 29 KJ%), which is different to other types of low-carbohydrate diets (i.e., ketogenic diet), where the majority of the calories is replaced by fat. Accordingly, the glucose-free diet used in our study did not result in an elevation of ketone bodies in the blood (Table S4). In patients with GSD III and GSD V (McArdle disease), a ketogenic diet could relieve symptoms, reduce CK levels and improve performance (Busch et al, 2005; Mayorandan et al, 2014; Francini-Pesenti et al, 2019). Therefore, future studies should investigate the effects of a ketogenic diet on the progression of dysferlinopathies.

Besides nutritional interventions, glycogen breakdown can be potently stimulated by exercise in a dynamic manner, concomitant with boosting mitochondrial function (Furrer et al, 2023a). We, therefore, tested whether an exercise intervention would also help normalize the metabolic phenotype of the dysferlinopathy animals by promoting repeated cycles of glycogen breakdown and synthesis. First, we observed that a single bout of treadmill running was sufficient to reduce muscle glycogen in the dysferlin-deficient muscles, in relative terms comparable with the reduction seen in WT mice (Fig 4A). Interestingly, inhibition and activation of enzymes involved in glycogen synthesis and breakdown are distinct between WT and Dysf −/− muscles (Table S2). First, protein levels of PYGM are only increased in Dysf −/− muscles after an acute bout of exercise to exhaustion (Fig S8A). Second, the phosphorylation of GYS1 and PYGM is affected differently upon exercise. For example, many phospho-sites of PYGM exhibit elevated phosphorylation after exercise in Dysf −/− mice; whereas, this is not the case in WT mice, in which some phospho-sites even display lower phosphorylation (Fig S8B and C). Despite the distinct activation of the enzymes, the net effect on glycogen breakdown appears to be similar. Importantly,

**Figure 3. Dysregulation of glucose metabolism in dysferlin-deficient muscles.**
**(A)** Relative protein abundance of glucose transporter, type 4 (GLUT4) calculated from the proteomic data in quadriceps muscle of 11 mo old mice and expressed relative to WT levels. **(B)** Glucose uptake in quadriceps muscle of 11 mo old mice 45 min after glucose (+2-Deoxyglucose) injection. **(C)** Blood glucose levels and area under the curve (AUC) during a glucose tolerance test (GTT) in 11 mo old mice. **(D, E, F)** Relative protein abundance (from proteomic data) of enzymes involved in glycolysis (D), (E) the rate limiting enzyme of the pentose phosphate pathway glucose-6-phosphate dehydrogenase (G6PD) and (F) glycogen synthase kinase 3 β (GSK3β), glycogen synthase (GYS1), and phosphorylase (PYGM) expressed relative to WT levels. **(G)** Representative examples of periodic acid-Schiff (PAS)–stained sections of quadriceps muscle of 11 mo old mice (scale bar = 200 $\mu$m). **(H)** Representative examples of electron microscopy images of quadriceps muscle of 11 mo old mice showing an accumulation of glycogen granules (large black dots, some of which are highlighted with a red arrow) in dysferlin-deficient muscle (scale bar = 1 $\mu$m). **(I, J, K)** Glycogen levels assessed biochemically in quadriceps muscle of mice at 4, 7 and 11 mo of age (I) or at the age of 11 mo in panels (J, K). **(L)** Relative protein abundance (from proteomic data) of GLUT4, GYS1 and PYGM in quadriceps muscles of two different dysferlinopathy mouse models at the age of 11 mo (expressed relative to WT levels). Data information: data represent means ± SEM. For panel (I), data are included from WT 4 mo n = 7, WT 7 mo n = 6, WT 11 mo n = 7, Dysf −/− 4 mo n = 7, Dysf −/− 7 mo n = 5, Dysf −/− 11 mo n = 6. **(A, B, C, D, E, F, I, J, K, L)** Empirical Bayes moderated t-statistics was used to analyze proteomic data and two-tailed *t* test (panel (B), AUC for the GTT in panel (C) and panel (K)), two-way repeated measures ANOVA (GTT panel (C)), two-way ANOVE with Šidák's multiple comparisons test (panel (I), asterisks indicate differences between the genotypes of the same age) or one-way ANOVA with Tukey's multiple comparisons test (panel (J)). *$P$ < 0.05; **$P$ < 0.01; ***$P$ < 0.001 (for panels (A, D, E, F, L), the asterisks represent *q*-values). GPI, glucose-6-phosphate isomerase; PFKM, phosphofructokinase, muscle; ALDOA, aldolase, fructose-bisphosphate A; TPI1, triosephosphate isomerase 1; GAPDH, glyceraldehyde-3-phosphate dehydrogenase; PGK1, phosphoglycerate kinase 1; PGAM2, phosphoglycerate mutase 2; ENO3, enolase 3; PKM, pyruvate kinase, muscle.

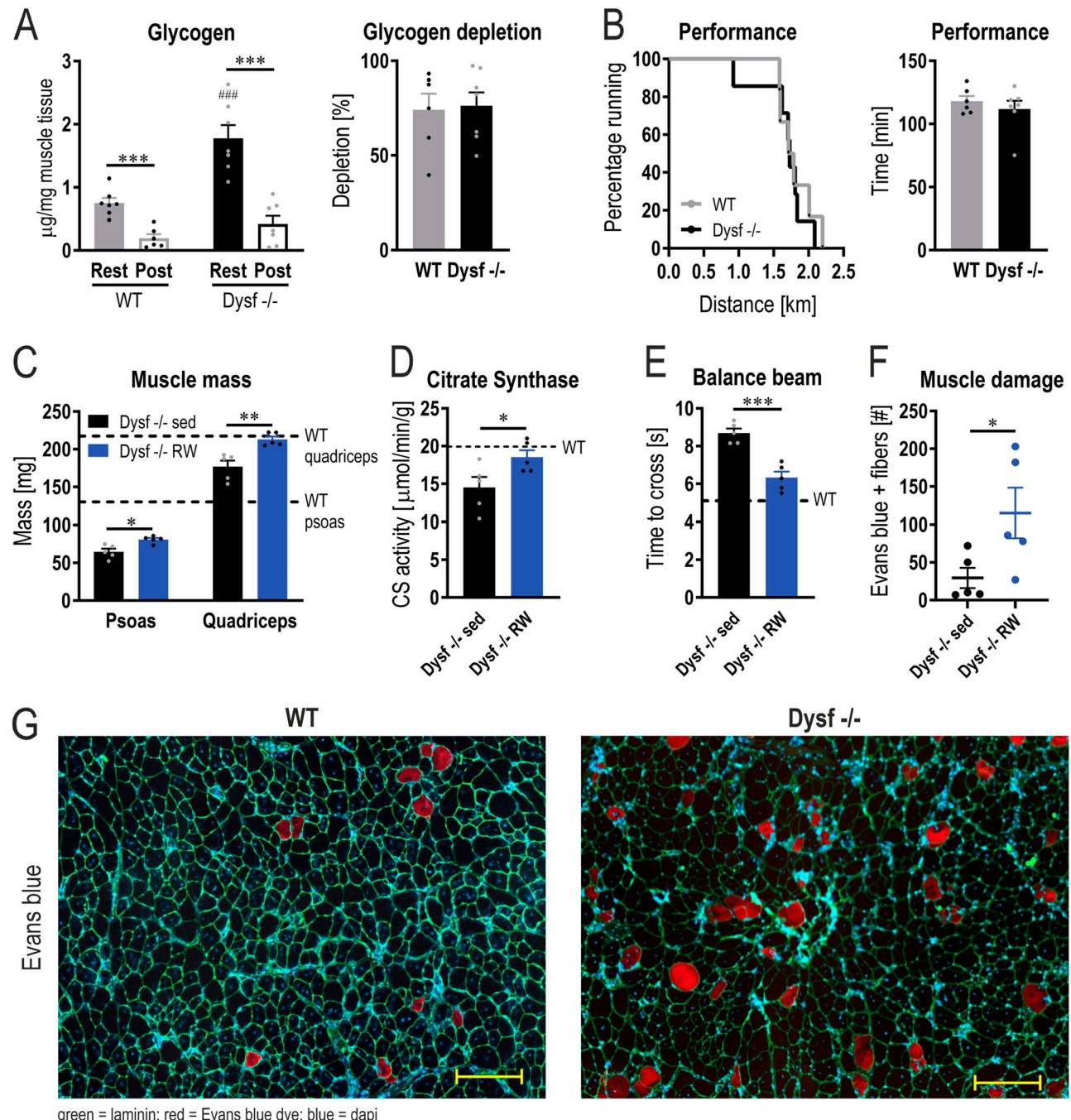

**Figure 4. Exercise ameliorates muscle atrophy and improves performance in dysferlin-deficient mice.**
**(A)** Glycogen levels assessed biochemically in quadriceps muscle of 11 mo old mice at rest and after one bout of exercise to exhaustion (post). Relative glycogen depletion was calculated using the value of mice at rest and those after exercise. **(B)** Maximal distance and time ran until mice reached exhaustion during a maximal performance test on a treadmill at the age of 11 mo. **(C)** Muscle mass at the age of 11 mo in sedentary Dysf −/− mice (Dysf −/− sed, black) and Dysf −/− mice after lifelong running wheel exercise (Dysf −/− RW, blue). Dashed lines represent the values of 11 mo old WT mice. **(D)** Citrate synthase (CS) activity in quadriceps muscles of 11 mo old mice. Dashed line represents the value of 11 mo old WT mice. **(E)** Performance on the balance beam at the age of 11 mo. Dashed line represents the value of 11 mo old WT mice. **(F)** Number of Evans blue-positive fibers in rectus femoris muscle of 11 mo old mice. **(G)** Representative examples of histological sections of rectus femoris muscle of 11 mo old mice that were injected with Evans blue dye (scale bar = 200 μm). Data information: data represent means ± SEM. **(A, B, C, D, E, F)** Asterisks indicate differences between the intervention groups (rest versus post-exercise for panel (A) within the same genotype or sed versus RW for panel (C, D, E, F)) using Log-rank (Mantel-Cox) test for the survival curve in panel (B) and two-tailed *t* test for all other comparisons. Hashtags indicate difference compared with the same group of the different genotype (WT rest versus Dysf −/− rest in panel (A)). *P < 0.05; **P < 0.01; ***P < 0.001.

glycogen depletion was the result of an exercise paradigm, in which similar performance and substrate usage were obtained in both phenotypes (Figs 4B and S9A–C). Moreover, AMP-dependent protein kinase (AMPK) phosphorylation and that of the downstream target acetyl-CoA-carboxylase (ACC), used as markers for metabolic stress, were comparable in WT and Dysf −/− in response to exercise (Fig S9D–H). This experiment established that exercise-induced changes in metabolic demand, including glycogen breakdown, are not impaired in dysferlin-deficient mice.

We then investigated whether increased dynamics of glycogen homeostasis and mitochondrial function by repeated exercise bouts can ameliorate disease progression. To address this question, dysferlin-deficient mice had free access to running wheels starting at the age of 2 mo and were compared with sedentary mice after 9 mo of training, thus at the age of 11 mo (Fig S10A). CK levels, body weight, and composition did not change with the exercise intervention (Fig S10B–D). In a homeostatic state at rest, thus unperturbed by the last bout of acute exercise, glycogen levels were not different between the two groups at the age of 11 mo (Fig S10E). Nevertheless, wheel running significantly increased muscle mass of psoas and quadriceps muscles and citrate synthase activity of quadriceps muscle (Fig 4C and D). In addition, balance beam performance was substantially improved (Fig 4E), suggesting that an exercise intervention is effective in counteracting muscle mass loss and ameliorating functional performance. However, even in the absence of notable changes in CK, a higher number of damaged muscle fibers were observed in the trained cohort (Fig 4F and G). This finding indicates that despite the therapeutic effect on mass and function, repeated bouts of exercise could have the potential to reduce myofiber integrity and increase damage, at least in the exercise protocol used here.

## Discussion

Dysferlin is a central mediator of vesicle-vesicle and vesicle-membrane fusion, and thus instrumental for damage mitigation, in particular in a mechanically challenged tissue such as skeletal muscle (Han & Campbell, 2007). However, in the past two decades, dysferlin has emerged as a multifaceted protein that is not only involved in membrane repair but also plays a role in the maintenance of t-tubule integrity, the regulation of excitation-contraction coupling and Ca$^{2+}$ handling (Quinn et al, 2024), Therefore, it might not be surprising that mini-dysferlin, which retains the capability to repair membranes, fails to ameliorate the dystrophy in preclinical models (Krahn et al, 2010; Lostal et al, 2012). These findings are substantiated when a dysferlinopathy patient was identified that harbor such a "mini-dysferlin" with membrane repair capacity but nevertheless exhibiting a muscle pathology (Krahn et al, 2010). This highlights the relevance of other cellular functions of dysferlin in the development of dystrophy. Our assessment of a potential therapeutic effect of muscle PGC-1α in dysferlinopathies, leading to an unexpected outcome, has resulted in the discovery of substantial remodeling of muscle metabolism in mice with an ablated dysferlin gene. Of note, this study was performed with male mice, and it is unclear whether all results also

apply to female mice. We demonstrate that muscles lacking dysferlin exhibit abnormal mitochondria and elevated death signaling. Furthermore, increased uptake of glucose observed in these muscles, followed by enhanced shunting into the pentose phosphate pathway and glycogen storage, most likely promoted by a throttling of glycolysis, indicate a shift in substrate usage and metabolism. This metabolic remodeling could thus contribute to the pathology of dysferlinopathies. First, excessive glycogen storage is an early event in etiology and is associated with the degree of which different muscles are affected, i.e., higher in myopathic proximal quadriceps muscle, and absent in distal TA muscle that does not exhibit muscle mass loss or other pronounced dystrophic features. It is noteworthy that for future studies, it is essential to consider the differences in disease manifestation between proximal and distal muscles. Whereas psoas is mostly affected and followed by the gluteus and quadriceps muscles, the proximal muscles (EDL, TA, and soleus) are completely protected from atrophy up to an advanced age and even exhibit compensatory hypertrophy. Second, the exacerbation of tissue pathology caused by elevated PGC-1α in some muscles aligns with the overlap in metabolic remodeling of glucose metabolism and accumulation of glycogen by PGC-1α and absence of dysferlin. Similar observations have been made in other disease models with high muscle glycogen, in which PGC-1α failed to promote an improvement or even worsened the outcome, e.g., Pompe disease (Takikita et al, 2010) or mice with an ablation of the mammalian target of rapamycin complex 1 (mTORC1) (Romanino et al, 2011). In contrast, a therapeutic effect of elevated muscle PGC-1α was reported in other wasting contexts, including Duchenne muscular dystrophy (Handschin et al, 2007), denervation-induced atrophy (Sandri et al, 2006), muscle damage induced by statins (Hanai et al, 2007), cardiotoxin-induced (Dinulovic et al, 2016a, 2016b) or crush injury (Haralampieva et al, 2018), sarcopenia (Garcia et al, 2018; Gill et al, 2018), amyotrophic lateral sclerosis (Da Cruz et al, 2012), cardiac cachexia (Geng et al, 2011), mitochondrial DNA mutator mice (Dillon et al, 2012), or hind limb unloading (Cannavino et al, 2014). Third, the PGC-1α-induced elevation in mitochondrial volume density and Ca$^{2+}$ concentration potentially exacerbates the pathology by triggering death signaling. Finally, endurance exercise resulted in a successful depletion of muscle glycogen, and when performed repeatedly, a significant amelioration of a number of morphological and functional parameters.

Based on our findings, it is evident that dysferlin plays a role in cell metabolism. It, however, is still unclear how the lack of dysferlin is linked to metabolic remodeling, and whether this is a primary or secondary process. At least some components, for example, GLUT4 availability in the cell membrane, could be linked to the function of dysferlin in vesicle trafficking and membrane fusion. Insulin-dependent glucose uptake is mediated by GLUT4 that is enriched in GLUT4 storage vesicles (GSVs) within the cytoplasm (Klip et al, 2019). Upon insulin stimulation, GLUT4 translocates to the plasma membrane via exocytosis of the GSVs, which is later on followed by the internalization of GLUT4 via endocytosis (Klip et al, 2019). Because dysferlin is involved in vesicle transport, dysferlin could also play a role in GLUT4 trafficking. For example, dysferlin has previously been shown to be engaged in endocytic recycling (Demonbreun et al, 2011). In particular, the internalization of the

transferrin receptor is delayed in dysferlin-deficient primary myoblasts (Demonbreun et al, 2011). Hence, it is conceivable that the internalization of various recycling cargo is dependent on the proper function of dysferlin. Accordingly, the internalization of GLUT4 could be slower in dysferlin-deficient muscles, which could result in an elevated glucose uptake. This process could be exacerbated by the dysregulation of Ca$^{2+}$ signaling in dysferlin-deficient muscles (Kerr et al, 2013, 2014; Muriel et al, 2022). In the absence of dysferlin, intracellular Ca$^{2+}$ levels are elevated for a prolonged period of time after laser injury compared with control myoblasts (Defour et al, 2014). Elevation of intracellular Ca$^{2+}$ by ionomycin treatment rapidly induces the translocation of GLUT4 to the plasma membrane (Li et al, 2014). Moreover, high levels of intracellular Ca$^{2+}$ reduce the internalization of GLUT4 (Li et al, 2014). Of note, an ionomycin-induced increase in intracellular Ca$^{2+}$ might be substantially higher than what is observed in dysferlin-deficient muscles after injury. Nevertheless, the higher Ca$^{2+}$ levels after muscle damage in dysferlin-deficient muscle could also contribute to the GLUT4 translocation and thereby glucose uptake.

The dysregulated Ca$^{2+}$ homeostasis in dysferlin-deficient muscle might also contribute to mitochondrial dysfunction and elevated death signaling. Of note, mitochondrial abnormalities are present in biopsies of dysferlinopathy patients (Sunitha et al, 2016). Similarly, a reduction in proteins involved in oxidative phosphorylation and a concomitant decrease in cellular respiration has been demonstrated in hiPSC-derived myobundles of patients (Khodabukus et al, 2024). This indicates that a mitochondrial phenotype exists in patients and could substantially affect cellular health. We now show that these mitochondrial abnormalities strongly promote death signaling, potentially driving the pathology. Whether these changes are triggered by a disruption of Ca$^{2+}$ homeostasis remains elusive. Strikingly, increasing Ca$^{2+}$ influx is sufficient to promote the development of muscular dystrophies and exacerbate the development of dysferlinopathy (Millay et al, 2009; Burr et al, 2014). Thus, targeting Ca$^{2+}$ dysregulation observed in many muscular dystrophies might provide therapeutic benefits across a broad range of diseases, including dysferlinopathies (Burr & Molkentin, 2015).

Finally, as a proof-of-concept, we provide evidence that promoting glycogen breakdown dynamically and restoring mitochondrial function by exercise can ameliorate muscle mass loss and performance. Because this running intervention is not feasible in patients because of the loss of independent ambulation, other forms of exercise or neuromuscular functional rehabilitation therapy programs could be implemented (D'Este et al, 2025). Importantly, an intervention should be designed to minimize the potential of muscle damage, e.g., by prioritizing concentric or isometric contractions, or use paradigms with minimal impact, such as swimming (Biondi et al, 2013). Moreover, potential damage should be continuously monitored with biomarkers, most importantly circulating CK. Furthermore, understanding the underlying mechanisms of these exercise-induced benefits is critical for developing strategies to target these pathways pharmacologically. It is noteworthy that increasing mitochondrial volume density alone might be detrimental because of an elevated potential to trigger death signaling, highlighting the importance of improving mitochondrial function in a coordinated and balanced manner. The powerful benefits of exercise most likely go beyond the effects on mitochondrial function and glycogen and may also affect other processes that are dysregulated in dysferlin-deficient muscles, such as lipid metabolism (Srivastava et al, 2017; Haynes et al, 2019).

In summary, our results imply the involvement of dysferlin in regulating myocellular metabolism, in particular that of glucose and glycogen and of mitochondria, and the subsequent triggering of death signaling (Fig 5). The substantial metabolic dysregulation in dysferlin-deficient muscles might contribute to the pathoetiology, thereby providing an explanation of the shortcomings of therapies exclusively aimed at restoring membrane repair. Further investigation should address whether dysferlin directly alters metabolism by interacting with GSVs, mitochondria or other structures, or whether some of the changes are a secondary effect triggered by Ca$^{2+}$ dysregulation. If substantiated in human patients, an amelioration of the metabolic dysregulation or Ca$^{2+}$ handling could, therefore, decisively help mitigate the pathology in combination with a mini-dysferlin. These findings thus not only shine light on hitherto unknown functions of dysferlin in the muscle cell but could help overcome current roadblocks in the treatment of dysferlinopathies.

# Materials and Methods

### Animals

All experiments were performed with male mice. The mouse strain B6.129-Dysf$^{tm1Kcam}$/J (stock no. 013149; Jackson Laboratory) was used as a model for dysferlin deficiency (Bansal et al, 2003; Wiktorowicz et al, 2015), and they were bread as homozygous colonies (Dysf −/−) in the Biozentrum. Mice were analyzed at the ages of 4 mo (~17 wk), 7 mo (~30 wk), 11 mo (~46 wk), and 15 mo (~65 wk) of age. Some of the findings were validated in 11-mo-old B6.A-Dysf$^{prmd}$/GeneJ (BLAJ) mice received from the Jain Foundation (stock no. 012767; Jackson Laboratory). B6.129-Dysf$^{tm1Kcam}$/J mice were also crossed with mice overexpressing PGC-1α specifically in muscle (PGC-1α mTG) to obtain Dysf −/− PGC-1α mice. The PGC-1α mTG animals were generated by crossing WT C57BL/6 mice with C57BL/6 mice expressing PGC-1α under the control of the muscle creatine kinase promoter as previously described (Lin et al, 2002). The littermates of the PGC-1α mTG breeding were used as WT control animals for all experiments. Mice were housed under standard conditions with a 12-h light:12-h dark cycle and had free access to water and standard rodent chow diet (3432-Maintenance; KLIBA NAFAG). To assess muscle damage by Evans blue dye incorporation, a 1% Evans blue dye (Sigma-Aldrich) solution (in 0.9% NaCl and sterile filtered) was administered intraperitoneally (i.p.) 16 h before the dissection at a dosage of 1% volume per gram of the mouse's body weight. Immediately after confirming death induced by CO$_2$ overdose, blood was collected from the vena cava in tubes containing lithium heparin (Sarstedt) and plasma prepared. Subsequently, the following hind limb muscles were freshly removed, weighted, snap-frozen in liquid nitrogen and stored at −80°C for further analysis: psoas, gluteus, quadriceps, TA, EDL, and soleus muscle. For mass spectrometry, immunoblotting, RT-qPCR, glucose

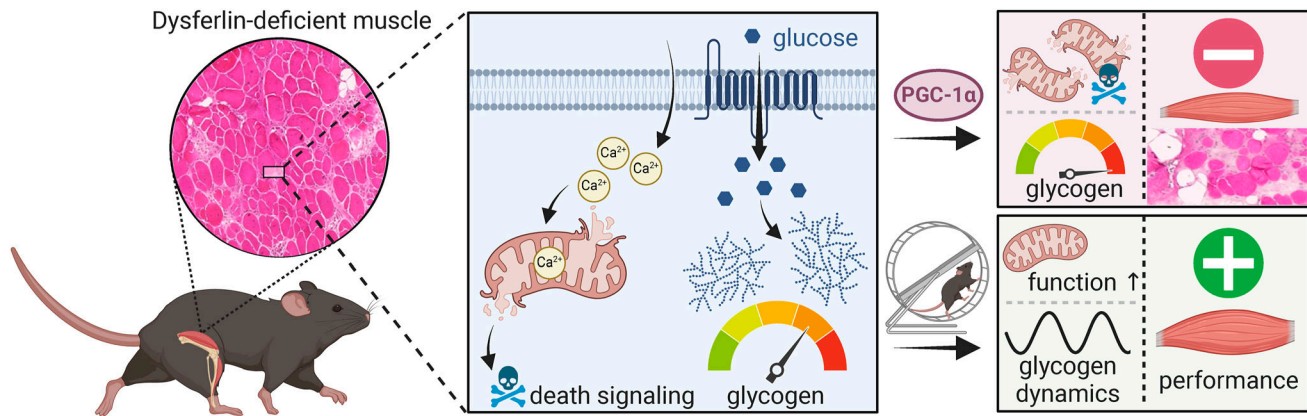

**Figure 5. Metabolic dysregulation in dysferlin-deficient muscle.**
Ca²⁺ homeostasis is dysregulated and mitochondria exhibit abnormalities in muscles lacking dysferlin. These mitochondrial abnormalities trigger death signaling. Furthermore, glucose uptake is elevated in the absence of dysferlin and redirected from glycolysis to glycogen synthesis resulting in an accumulation of glycogen. The increase in mitochondrial volume density and muscle glycogen by overexpressing PGC-1α (peroxisome proliferator-activated receptor γ coactivator 1α) in muscle exacerbates disease progression whereas improving mitochondrial function and stimulating the dynamics of glucose and glycogen metabolism by exercise ameliorates muscle mass loss and improves performance. Created with BioRender.com, with permission.

uptake, glycogen content, Ca²⁺ concentration, citrate synthase activity, and mitochondrial respiration (using Seahorse) measurements, the quadriceps muscle was pulverized before further processing. Hence, the muscle tissue homogenate contained a proportional mixture of all abundant fiber types. The quadriceps muscle of the other limb was, in addition, embedded in 7% tragacanth (Sigma-Aldrich), snap-frozen in liquid nitrogen-cooled isopentane (−150°C), and stored at −80°C. All experiments adhered to the Swiss guidelines for animal experimentation and care and were approved by the Kantonales Veterinäramt Basel-Stadt.

## Interventions to modulating muscle glycogen

### Dietary intervention
Mice of the intervention group receive an ad libitum glucose-free/low-carbohydrate diet (32 kJ% fat, 66 kJ% protein, 2 kJ% carbohydrates; ssniff Spezialdiäten GmbH) from the age of 3 mo until 11 mo of age. Control animals received a standard chow diet. Two months after the intervention, blood levels of the keton body β-hydroxybutyrate (β-OHB) were measured in WT mice using a β-OHB meter (Precision Xtra, Abbott Laboratories). At the end of the experiment, blood glucose was assessed Dysf −/− mice with a glucose meter (Accu-Check, Roche Diagnostics).

### Exercise intervention
The lifelong exercise intervention consisted of free running wheel access. Mice were single caged and received running wheels (Columbus Instruments) from the age of 2 mo until 11 mo. Voluntary wheel running activity was monitored throughout the intervention. Creatine kinase (CK) levels were measured at the ages of 7 and 11 mo and balance beam performance and body composition at the end of the intervention at the age of 11 mo. Furthermore, Evans blue dye solution was injected intraperitoneally (i.p.) 16 h before the dissection. Running wheels were removed 1 d before the termination of the experiment.

## In vivo measurements

### Body composition
Body composition (including lean and fat mass) was assessed in restrained, conscious mice using EchoMRI-100 analyzer (EchoMRI Medical Systems) and expressed relative to body mass.

### Grip strength
An estimate of the muscle force of the fore and hind limbs together was obtained by measuring grip strength with a Grip Strength Meter (Chatillon, Columbus Instruments). The mouse was positioned in a way that all four limbs rested on the inclined mesh pull bar, which was linked to a force transducer. Subsequently, the mouse was pulled horizontally away from the sensor at a uniform speed until it released its grip, ensuring consistent testing conditions. During the testing session, a series of three consecutive pulls were executed, and the highest value was recorded in kilogram-force. This procedure was repeated three times with a minimal rest of 10 min between the tests. The two highest values were averaged and expressed relative to body weight.

### Voluntary activity
Voluntary activity was assessed using running wheels (Columbus Instruments). Mice received free access to running wheels for 3 d that monitored voluntary activity. The 1st d was used to acclimatize the mice and the activity of the last 48 h was analyzed and averaged. To identify "non-runners," voluntary activity was tested in 2-mo-old mice. Mice that ran less than 1 km in 48 h at the age of 2 mo were excluded from the analysis at the age of 6 and 10 mo.

### Maximal exercise performance
Maximal performance was assessed on a closed motorized Metabolic Modular Treadmill system (Columbus Instruments) that allows the measurement of VO₂ and VCO₂. Before the test, mice were familiarized with treadmill running on 2 d for 15–20 min at velocities from 5 to 12 m/min. The maximal performance was tested at an

inclination of 5° and a protocol that progressively increased speed as follows: 5 min at 5 m/min, 5 min at 8 m/min, 15 min at 10 m/min, followed by an increase of 2 m/min every 15 min until exhaustion. Exhaustion was determined when the mice could no longer maintain the pace, despite the mild electric stimulus at the end of the treadmill. Blood lactate levels were assessed pre- and post-exercise using a lactate meter (Nova Biomedical). Distance, time, and $VO_{2max}$, and respiratory exchange ratio (RER) were determined. To determine post-exercise glycogen levels, mice were immediately euthanized, tissue collected, snap-frozen in liquid nitrogen, and stored at –80°C.

### Balance beam

Balance was tested on a 1 cm-wide beam with an inclination of 9°, where the mice had to voluntarily cross the 80 cm distance as fast as possible. The test was performed in the dark with a bright light shining at the starting platform as an adverse stimulus. At the end of the beam there was a red Plexiglas house where the mouse could rest at the end of the test. After 1 d of acclimatization (including three runs), mice were tested on three consecutive days. On each day, mice performed three runs across the beam and time to cross was measured. The average time of the fastest two runs (of all nine runs) was calculated.

### Gait analysis

Gait was assessed using the Noldus CatWalk XT system as previously described (Hamers et al, 2001) and following the CatWalk XT 10.6 Reference Manual. Mice were placed at one end of the glass plate and given the freedom to move back and forth along the walkway. Only runs where mice walked continuously and straight across the walkway were considered successful. Additionally, a run had to adhere to the following criteria: run duration between 0.5 and 5 s with a maximum allowed speed variation of 60%. We established a requirement for a minimum of three valid runs to be completed. The CatWalk XT software was used to analyze the runs and to assess various gait parameters, such as run speed (cm/s), cadence (number of steps/s), stride length (cm), and stand and swing time (s). The fastest run of the acquired successful runs was considered for the final analysis.

### GTT

After an overnight fast (16 h), mice were weighted and basal glucose levels determined from the tail vein using a glucose meter (Accu-Check, Roche Diagnostics). Subsequently, 2 g of glucose (Sigma-Aldrich) per kg body weight was i.p.-injected and glucose levels were measured 15, 30, 45, 60, 90, 120, and 180 min after glucose injection. A 20% glucose solution diluted in 0.9% NaCl was used.

### Plasma analysis

To measure CK levels, blood was either collected from the vena cava at the end of the experiments or from the tail vein for longitudinal assessment (only during the exercise intervention) in tubes containing lithium heparin (Sarstedt) and plasma was obtained after centrifuging at 2,000$g$ for 5 min. CK levels were determined using the cobas c 111 analyzer (cobas, Roche Diagnostics AG). For the assessment of plasma insulin, blood was collected from the tail

vein in EDTA-coated tubes (Sarstedt) after overnight fasting and 5 and 25 min after i.p.-injection of glucose (2 g glucose per kg body weight), and plasma was obtained after centrifuging at 2,000$g$ for 5 min. Insulin levels were determined using the MSD Mouse/Rat Insulin Kit (K152BZC; Meso Scale Discovery) according to the manufacturer's protocol.

### Citrate synthase (CS) activity

To assess CS activity, 6–10 mg of pulverized muscle was homogenized in a 100 $\mu$l/mg homogenizing solution containing 0.1 M $KH_2PO_4$ and 0.05% BSA (pH 7.3) as previously described (Srere, 1969; Perry et al, 2010). 10 $\mu$l of the muscle homogenate was used for the analysis. The following reagents were added to the muscle homogenate: 150 $\mu$l 100 mM TRIS buffer (pH 8.3), 25 $\mu$l 1 mM DTNB 5,5′-Dithiobis(2-nitrobenzoic acid), 40 $\mu$l 3 mM Acetyl-CoA and 10 $\mu$l Triton (10%). Before the measurement, 15 $\mu$l of 10 mM oxaloacetate was added, and following mixing, the kinetics was measured at 412 nm at 20 s intervals for 5 min (at 37°C). The change in absorption within the first minute was used for the analysis.

### Mitochondrial respiration (Seahorse)

Mitochondrial respiration was measured from frozen tissue using XF96 Seahorse as previously described (Osto et al, 2020). Briefly, 10 mg of pulverized muscle was homogenized in MAS buffer containing 70 mM sucrose, 220 mM mannitol, 5 mM $KH_2PO_4$, 5 mM $MgCl_2$, 1 mM EGTA and 2 mM HEPES (pH adjusted to 7.4). Because TCA cycle function is disrupted and mitochondria are uncoupled because of freezing and thawing (Acin-Perez et al, 2020; Osto et al, 2020), we only assessed respiration of complex II and IV. The following injections were performed: port A: 5 mM succinate + 2 $\mu$M rotenone; port B: 4 $\mu$M antimycin A; port C: 0.5 mM TMPD (N,N,N′,N′-Tetramethyl-p-phenylenediamine dihydrochloride) + 1 mM ascorbic acid (pH 7.4); port D: 50 mM sodium azide. The cytochrome c concentration in the MAS buffer used for the assay was 100 $\mu$g/ml. We loaded 6 $\mu$g of protein per sample and measured 6 biological and five technical replicates. If the well did not respond to an injection, the technical replicate was removed from the analysis. If more than two technical replicates had to be removed from a biological replicate, the entire biological replicate was not considered for the analysis. Of the 120 tested samples (24 samples × 5 technical replicates), one biological and eight technical replicates (all belonging to different biological replicates) had to be removed. Complex II and IV were calculated by subtracting respiration measured after inhibiting complex II or IV from the actual complex II and IV measurements (first measurement after injection two or four minus first measurement after injection one or three). The data are visualized as a relative change in response to the injections.

### Total Ca$^{2+}$ concentration

Total Ca$^{2+}$ concentration was measured using the colorimetric Calcium Assay Kit (ab102505) according to the manufacturer's instruction. Briefly, 40 mg of pulverized muscle was homogenized in 200 $\mu$l (5x) Calcium Assay Buffer. In the reaction wells, 50 $\mu$l of undiluted homogenate was used.

## Glucose uptake and muscle glycogen

To measure glucose uptake in muscle, mice were i.p.-injected with 100 mg/kg 2-Deoxyglucose (2DG; Sigma-Aldrich) and 1.9 g/kg glucose (Sigma-Aldrich) after an overnight fast. Muscle tissue was collected after 45 min, snap-frozen in liquid nitrogen, and stored until further analysis in –80°C. Glucose uptake was assessed with the 2-Deoxyglucose (2DG) Uptake Measurement Kit (CSR-OKP-PMG-K01; Cosmo Bio LTD) following the manufacturer's protocol. Muscle glycogen was measured with the fluorometric assay of the Glycogen Assay Kit (ab65620; Abcam) according to the manufacturer's instructions.

## Histology and immunofluorescence staining

The quadriceps muscle was embedded in 7% tragacanth (Sigma-Aldrich), snap-frozen in liquid nitrogen-cooled isopentane (–150°C) and stored at –80°C. Serial sections of 10 $\mu$m were cut from the mid-belly with the cryostat (CM1950; Leica) at a temperature of –21°C. The sections were collected on SuperFrost Plus adhesion slides (Menzel Gläser, Thermo Fisher Scientific) and stored at –80°C until further processed.

Before the hematoxylin and eosin (H&E; MHS32, HT110232; Sigma-Aldrich) and periodic acid-Schiff (PAS; 395B-1KT; Sigma-Aldrich) staining, dried sections were fixed with 4% PFA. The stainings were performed according to the manufacturer's protocols. H&E-stained sections were dehydrated with 75%, 90%, and 100% EtOH followed by 100% xylene and mounted with Eukitt mounting medium (Kindler GmbH); whereas, the PAS-stained sections were dehydrated with 70%, 95%, and 100% EtOH, followed by 100% xylene and mounted with Histomount Mounting Solution (Invitrogen). Oil Red O (ORO, O0625; Sigma-Aldrich) staining was performed on dried sections as previously described (Mehlem et al, 2013). The ORO staining was performed for 15 min at RT and Mount Quick aqueous mounting medium (Bio-Optica) was used.

The sections used to analyze Evans blue dye incorporation were, in addition, stained for laminin. Before the staining, the sections were rehydrated in phosphate-buffered saline (PBS) and blocked in PBS containing 0.4% Triton X-100 (Sigma-Aldrich) and 10% goat serum (Sigma-Aldrich). Subsequently, sections were incubated for 1 h with a primary antibody solution containing Anti-Laminin antibody (ab11575; Abcam) and 10% goat serum. After washing the sections three times with PBS, they were incubated with a secondary antibody solution containing Goat anti-Rabbit IgG Alexa Fluor 488 (A-11008; Invitrogen) and 10% goat serum. After washing with PBS and dehydrate in 70% and 100% EtOH, sections were mounted with ProLong Gold Antifade Reagent with DAPI (Invitrogen). Muscle sections were imaged with the Axio Scan.Z1 Slide Scanner (Zeiss). To assess muscle damage, the Evans blue-positive muscle fibers of the rectus femoris of the quadriceps muscle were manually counted.

## Resin embedding and thin sectioning for transmission electron microscopy (TEM)

Quadriceps muscle was removed and small pieces of ~1 × 1 × 4 mm were cut carefully without squeezing the tissue. Subsequently, the muscle pieces were fixed overnight at 4°C in 0.1 M PIPES buffer with 2 mM $CaCl_2$, 3% PFA and 2.5% glutaraldehyde. Fixed samples are washed once in PBS buffer and post-fixed first in 1% reduced osmium tetroxide (containing 1.5% potassium ferricyanide) for 40 min and subsequently in 1% osmium tetroxide for 40 min. After washing in $H_2O$, fixed samples were dehydrated in ascending EtOH grades 25%, 50%, 75%, 90%, and 100%. Dehydrated samples are incubated in Epon resin/EtOH mixture (50/50) for 30 min and then embedded overnight in Epon resin at RT. The samples are transferred in mould filled with fresh resin and incubated at 60°C for 48 h. Thin sections of 60 nm were cut using an ultramicrotome UC7 (Leica) transferred on EM grids and subsequently stained with 2% uranyl acetate and 3% lead citrate. EM micrographs were recorded on Veleta CCD camera (Olympus) using a T12 Spirit transmission electron microscope (FEI) operating at an acceleration voltage of 80 kV. To visualize glycogen accumulation, cross-sectional sections were imaged. For the quantification of mitochondrial morphology, longitudinal sections were imaged with a magnification of 13,000x.

To quantify mitochondrial morphology, all images were opened and visualized using Napari (Sofroniew et al, 2024), a multi-dimensional image viewer for Python. Mitochondrial segmentation was performed using empenada-napari (Conrad et al, 2022), a plugin for Napari that provides a suite of tools for image segmentation. Initially, automatic segmentation was performed using the plugin's built-in deep learning algorithm MitoNet (Conrad & Narayan, 2023) to identify mitochondrial structures. However, because of the complexity of the images and the variability in mitochondrial morphology, manual correction of the automatic segmentation was often necessary to ensure accurate results. This was achieved using Napari-Empanada's interactive tools, which allow users to manually edit and refine the segmentation masks. By combining the automatic segmentation capabilities of Napari-Empanada with manual correction, we were able to achieve high-quality segmentations of mitochondria in our images. Measurements were then extracted using napari-skimage-regionprops (Haase et al, 2023), a plugin allowing to extract quantitative features from the segmented mitochondria. By combining the automatic segmentation capabilities of Napari-Empanada with manual correction and quantitative measurements using Napari-skimage-regionprops, we were able to achieve high-quality segmentations and extract meaningful metrics from our images. At least five images per sample were analyzed or the number of images needed to include measurements of at least 100 mitochondria. Three biological replicates were included per group. Mitochondrial volume density and number of mitochondria were calculated per image (average of at least five images); whereas, the size was calculated from at least 100 measured mitochondria.

## Immunoblotting

Protein isolation and immunoblotting was performed as previously described (Perez-Schindler et al, 2012; Leuchtmann et al, 2023). Briefly, 25–30 mg of pulverized muscle was homogenized in 300 $\mu$l ice-cold radio-immunoprecipitation assay (RIPA) buffer (150 mM NaCl, 1% vol/vol Nonidet-P40 substitute, 0.2% vol/vol Na-deoxycholate, 0.1% vol/vol SDS, 50 mM TRIS–HCl [pH7.5], 1 mM EDTA, 1 mM DTT, 10 mM nicotinamide) containing protease

**Life Science Alliance**

inhibitors (cComplete Mini EDTA free, Roche) and phosphatase inhibitors (PhosSTOP, Roche) and protein concentration was determined by BCA Protein Assay Kit (Thermo Fisher Scientific). Subsequently, samples were diluted and denatured in 4X Laemmli Sample Buffer, 45 μg of protein was separated on a 10% SDS-polyacrylamide gel and transferred to a nitrocellulose membrane (0.45 NC, 10600007, 0.45 μm; Amersham Protran). After verifying protein transfer by Ponceau S staining, the membrane was blocked for 1 h in 5% BSA and incubated overnight at 4°C with primary antibodies. The following antibodies were used: TBK1/NAK (D1B4) (#3504; dilution 1:500; Cell Signaling), cGAS (D3O8O) (#31659; dilution 1:1,000; Cell Signaling) and STING (D1V5L) (#50494; dilution 1:1,000; Cell Signaling). After washing, the membrane was incubated with HRP-linked Antibody (#7074; dilution 1:3,000; Cell Signaling) for 1 h at RT. The following chemoluminescent substrates were used for detection on the ChemiDoc Imaging System (Bio-Rad): SuperSignal West Dura (#34076) or Femto (#34095); Pierce. Subsequently, the membrane was washed and stripped for 20 min at RT using Restore PLUS Western blot Stripping Buffer (#46430; Thermo Fisher Scientific). The procedure including blocking, overnight primary antibody incubation, secondary antibody incubation and detection was repeated using the following primary antibodies: Phospho-TBK1/NAK (Ser172) (D52C2) (#5483; dilution 1:500; Cell Signaling) and Phospho-STING (Ser365) (D8F4W) (#72971; dilution 1:1,000; Cell Signaling).

### RNA extraction and RT-qPCR

After pulverization of the muscle, tissue was homogenized in 1 ml TRIzol agent (Sigma-Aldrich) with FastPrep tubes (MP Biomedicals) and RNA isolated following to the manufacturer's protocol. RNA concentration and quality were measured using the NanoDrop OneC spectrophotometer (Thermo Fisher Scientific). Next, 1 μg of RNA was subjected to deoxyribonuclease I treatment (Invitrogen) and then reverse-transcribed using the High Capacity cDNA RT Kit (Applied Biosystems). Gene expression was assessed using the QuantStudio 5 Real-Time PCR System (Applied Biosystems) with the Fast SYBR Green Master Mix (Applied Biosystems). The ct values were normalized to the housekeeping gene TATA box–binding protein (*Tbp*) and gene expression was presented relative to the WT controls using the $2^{-\Delta\Delta Ct}$ methodology. *Tbp* was found to be a suitable housekeeping genes, with mean ct values of 25.569 for WT (range 25.266–25.996), 25.621 for Dysf –/– (range 25.303–26.168), 25.621 for PGC-1α mTG (range 25.305–25.824), and 26.599 for Dysf –/– PGC-1α (range 26.229–27.066). Only Dysf –/– PGC-1α was different to the other groups, but even when we tested *S18* as a housekeeper, the ct value of the Dysf –/– PGC-1α group was also 1 cycle later compared with that of the other three groups that were almost identical (ct of ~12.2 in Dysf –/– PGC-1α versus ~11.2 in other samples), suggesting that the cDNA concentration is lower in the samples of the Dysf –/– PGC-1α group. Following primer sequences were used (5'–3'): *Agl* (glycogen debranching enzyme): forward—CATGAAGGACGAGGGTTTCA, reverse—TTAAAAGTGCCTCGGCA-GAC; *Gaa* (α-1,4-glucosidase): forward—ACGCGGCATAGGCCTTC, reverse—GCAGCATTAACTCCCGAAGC; *Ppargc1a* (gene encoding for PGC-1α) forward—AGCCGTGACCACTGACAACGAG, reverse—GCTGCATG GTTCTGAGTGCTAAG; *Pygm* (glycogen phosphorylase) forward—GGT

TTATGGTGCCGAGGACT, reverse—GGCGGCGGGAATAACTTTCT; *Tbp*: forward—TGCTGTTGGTGATTGTTGGT, reverse—CTGGCTTGTGTGGGAAA GAT.

### Mass spectrometry analysis

Sample preparation for proteomic and phosphoproteomic analysis (including phosphopeptide enrichment) of the following samples: WT, Dysf –/–, PGC-1α mTG, Dysf –/– PGC-1α, WT exercise, and Dysf –/– exercise.

Muscle issue was pulverized and 10 mg lysed in 8 M urea, 0.1 M ammonium bicarbonate, phosphatase inhibitors (Sigma-Aldrich) by sonication (Bioruptor, 10 cycles, 30 s on/off; Diagenode), and proteins were digested as described previously (Ahrne et al, 2016). Shortly, proteins were reduced with 5 mM TCEP for 60 min at 37°C and alkylated with 10 mM chloroacetamide for 30 min at 37°C. After diluting samples with 100 mM ammonium bicarbonate buffer to a final urea concentration of 1.6 M, proteins were digested by incubation with sequencing-grade modified trypsin (1/50, w/w; Promega) for 12 h at 37°C. After acidification using 5% TFA, peptides were desalted using C18 reverse-phase spin columns (Macrospin, Harvard Apparatus) according to the manufacturer's instructions, dried under vacuum, and stored at –20°C until further use.

For phosphoproteomic analysis, 300 μg protein were digested and resulting peptides were enriched for phosphorylated peptides using Fe(III)-IMAC cartridges on an AssayMAP Bravo platform as recently described (Post et al, 2017).

#### *LC–MS/MS analysis—proteomics*

Dried peptides were resuspended in 0.1% aqueous formic acid and subjected to LC–MS/MS analysis using a Orbitrap Fusion Lumos Mass Spectrometer fitted with an EASY-nLC 1,200 (both Thermo Fisher Scientific) and a custom-made column heater set to 60°C. Per sample, 250 ng of peptide was resolved using a RP-HPLC column (75 μm × 36 cm) packed in-house with C18 resin (ReproSil-Pur C18–AQ, 1.9 μm resin; Dr. Maisch GmbH) at a flow rate of 0.2 μl/min. The following gradient was used for peptide separation: from 5% B to 12% B over 5 min to 35% B over 65 min to 50% B over 20 min to 95% B over 2 min followed by 18 min at 95% B. The runtime per sample including column equilibration, sample loading and gradient separation of peptides was ~140 min. Buffer A was 0.1% formic acid in H2O and buffer B was 80% acetonitrile, 0.1% formic acid in H2O.

The mass spectrometer was operated in DDA mode with a cycle time of 3 s between master scans. Each master scan was acquired in the Orbitrap at a resolution of 240,000 FWHM (at 200 m/z) and a scan range from 375 to 1,600 m/z followed by MS2 scans of the most intense precursors in the linear ion trap at "Rapid" scan rate with isolation width of the quadrupole set to 1.4 m/z. Maximum ion injection time was set to 50 ms (MS1) and 35 ms (MS2) with an AGC target set to $1 \times 10^6$ and $1 \times 10^4$, respectively. Only peptides with charge state 2–5 were included in the analysis. Monoisotopic precursor selection (MIPS) was set to peptide, and the intensity threshold was set to $5 \times 10^3$. Peptides were fragmented by higher-energy collisional dissociation (HCD) with collision energy set to

35%, and one microscan was acquired for each spectrum. The dynamic exclusion duration was set to 30 s.

### LC–MS/MS analysis—phosphoproteomics

Phospho-enriched peptides were resuspended in 0.1% aqueous formic acid and subjected to LC–MS/MS analysis using a Orbitrap Fusion Lumos Mass Spectrometer fitted with an EASY-nLC 1,200 (both Thermo Fisher Scientific) and a custom-made column heater set to 60°C. Per sample, 15% of the phospho-enriched peptide preparation was resolved using a RP-HPLC column (75 $\mu$m × 36 cm) packed in-house with C18 resin (ReproSil-Pur C18–AQ, 1.9 $\mu$m resin; Dr. Maisch GmbH) at a flow rate of 0.2 $\mu$l/min. The following gradient was used for peptide separation: from 5% B to 8% B over 5 min to 20% B over 45 min to 25% B over 15 min to 30% B over 10 min to 35% B over 7 min to 42% B over 5 min to 50% B over 3 min to 95% B over 2 min followed by 18 min at 95% B. The runtime per sample including column equilibration, sample loading and gradient separation of peptides was ~130 min. Buffer A was 0.1% formic acid in water and buffer B was 80% acetonitrile, 0.1% formic acid in water.

The mass spectrometer was operated in DDA mode with a cycle time of 3 s between master scans. Each master scan was acquired in the Orbitrap at a resolution of 120,000 FWHM (at 200 m/z) and a scan range from 375 to 1,600 m/z followed by MS2 scans of the most intense precursors in the Orbitrap at a resolution of 30,000 FWHM (at 200 m/z) with isolation width of the quadrupole set to 1.4 m/z. Maximum ion injection time was set to 50 ms (MS1) and 54 ms (MS2) with an AGC target set to $1 \times 10^6$ and $5 \times 10^4$, respectively. Only peptides with charge state 2–5 were included in the analysis. MIPS was set to peptide, and the intensity threshold was set to $2.5 \times 10^4$. Peptides were fragmented by HCD with collision energy set to 30%, and one microscan was acquired for each spectrum. The dynamic exclusion duration was set to 30 s.

### Data analysis—proteomics

The acquired raw-files were imported into the Progenesis QI software (v2.0, Nonlinear Dynamics Limited), which was used to extract peptide precursor ion intensities across all samples applying the default parameters. The generated mgf-file was searched using MASCOT against a murine database (consisting of 17,013 protein sequences downloaded from Uniprot on 20190307) and 392 commonly observed contaminants using the following search criteria: full tryptic specificity was required (cleavage after lysine or arginine residues, unless followed by proline); three missed cleavages were allowed; carbamidomethylation (C) was set as fixed modification; oxidation (M) and acetyl (Protein N-term) were applied as variable modifications; mass tolerance of 10 ppm (precursor) and 0.6 D (fragments). The database search results were filtered to set the false discovery rate (FDR) to 1% on the peptide and protein level, respectively. Quantitative analysis results from label-free quantification were processed using the SafeQuant R package v.2.3.2. (https://github.com/eahrne/SafeQuant/) (Ahrne et al, 2016) to obtain peptide relative abundances. This analysis included global data normalization by equalizing the total peak/reporter areas across all LC-MS runs, data imputation using the knn algorithm, summation of peak areas per protein and LC–MS/MS run, followed by calculation of peptide abundance ratios and testing for differential abundance using empirical Bayes moderated t-statistics (as implemented in the R/Bioconductor limma package). The resulting per protein and condition comparison P-values were adjusted for multiple testing using the Benjamini-Hochberg method. To meet additional assumptions (normality and homoscedasticity) underlying the use of linear regression models and t Tests, MS-intensity signals were transformed from the linear to the log-scale. Data are presented in Table S1.

In all proteomic analyses, only proteins with more than one peptide were taken into account. Furthermore, we applied a $\log_2$ fold change (FC) threshold of ±0.2 (proteins with a $\log_2$(FC) > 0.199 or <−0.199) and only considered q-value < 0.01 statistically significant for the volcano plot and functional annotation analysis. To determine functional annotation clusters of gene ontology (GO) biological processes and REACTOME pathways, the Database for Annotation, Visualization and Integrated Discovery (DAVID; https://david.ncifcrf.gov/tools.jsp) platform was used (Huang et al, 2009; Sherman et al, 2022). A background list was used including all detected proteins with more than one peptide and functional annotation clusters with an enrichment score >2 were considered. For the visualization of different proteins, we calculated the fold change relative to the WT group (if not otherwise stated) using the measured intensities. We considered q-value < 0.05 as statistically significant and indicated this with asterisks in the figures.

### Data analysis—phoshoproteomics

The acquired raw-files were imported into the Progenesis QI software (v2.0, Nonlinear Dynamics Limited), which was used to extract peptide precursor ion intensities across all samples applying the default parameters. The generated mgf-file was searched using MASCOT against a murine database (consisting of 34,026 forward and reverse protein sequences downloaded from Uniprot on 20190307) and 392 commonly observed contaminants using the following search criteria: full tryptic specificity was required (cleavage after lysine or arginine residues, unless followed by proline); three missed cleavages were allowed; carbamidomethylation (C) was set as fixed modification; oxidation (M) and phosphorylation (STY) were applied as variable modifications; mass tolerance of 10 ppm (precursor) and 0.02 D (fragments). The database search results were filtered using the ion score to set the false discovery rate (FDR) to 1% on the peptide and protein level, respectively, based on the number of reverse protein sequence hits in the datasets. Quantitative analysis results from label-free quantification were processed using the SafeQuant R package v.2.3.2. (https://github.com/eahrne/SafeQuant/) (Ahrne et al, 2016) to obtain peptide relative abundances. This analysis included global data normalization by equalizing the total peak/reporter areas across all LC-MS runs, data imputation using the knn algorithm, summation of peak areas per, and LC–MS/MS run, followed by calculation of peptide abundance ratios. Only isoform specific peptide ion signals were considered for quantification. To meet additional assumptions (normality and homoscedasticity) underlying the use of linear regression models and t tests, MS-intensity signals were transformed from the linear to the log-scale. The summarized peptide expression values were used for statistical testing of between condition differentially abundant peptides. Here, empirical Bayes moderated t tests were applied, as implemented in the R/Bioconductor limma package (http://bioconductor.org/packages/release/bioc/html/limma.html)

were used. The resulting per protein and condition comparison *P*-values were adjusted for multiple testing using the Benjamini-Hochberg method. Data are presented in Table S2. Because some modifications were measured multiple times and some peptides included multiple modifications, we summed up all the intensities of one modification. If one peptide contained two modifications, the intensity was spitted equally among the two modifications. The statistics of the shown phosphorylation changes was performed using a two-tailed *t* test and *P* < 0.05 was considered statistical significant. For the visualization, we calculated the fold change relative to the WT group or "rest" group (e.g., WT rest or Dysf −/− rest) using the calculated intensities.

### Validation mass spectrometry experiment of WT, Dysf −/−, and BLAJ mice

10 mg of pulverized tissue were resuspended in lysis buffer (5% SDS, 10 mM TCEP, 0.1 M TEAB) and lysed by sonication using a PIXUL Multi-Sample Sonicator (Active Motif) with Pulse set to 50, PRF to 1, Process Time to 10 min and Burst Rate to 20 Hz. Lysates were incubated for 10 min at 95°C, alkylated in 20 mM iodoacetamide for 30 min at 25°C, and proteins digested using S-Trap micro spin columns (Protifi) according to the manufacturer's instructions. Shortly, 12% phosphoric acid was added to each sample (final concentration of phosphoric acid 1.2%) followed by the addition of S-trap buffer (90% methanol, 100 mM TEAB pH 7.1) at a ratio of 6:1. Samples were mixed by vortexing and loaded onto S-trap columns by centrifugation at 4,000*g* for 1 min followed by three washes with S-trap buffer. Digestion buffer (50 mM TEAB pH 8.0) containing sequencing-grade modified trypsin (1/25, w/w; Promega) was added to the S-trap column and incubate for 1 h at 47°C. Peptides were eluted by the consecutive addition and collection by centrifugation at 4,000*g* for 1 min of 40 *μ*l digestion buffer, 40 *μ*l of 0.2% formic acid, and finally, 35 *μ*l 50% acetonitrile, 0.2% formic acid. Samples were dried under vacuum and stored at −20°C until further use.

Dried peptides were resuspended in 0.1% aqueous formic acid and subjected to LC–MS/MS analysis using a timsTOF Ultra Mass Spectrometer (Bruker) equipped with a CaptiveSpray nano-electrospray ion source (Bruker) and fitted with a Vanquish Neo (Thermo Fisher Scientific). Per sample, 50 ng of peptide was resolved using a RP-HPLC column (100 *μ*m × 30 cm) packed in-house with C18 resin (ReproSil Saphir 100 C18, 1.5 *μ*m resin; Dr. Maisch GmbH) at a flow rate of 0.4 *μ*l/min and column heater set to 60°C. The following gradient was used for peptide separation: from 2% B to 25% B over 25 min to 35% B over 5 min to 95% B over 1 min followed by 5 min at 95% B to 2% B over 1 min followed by 3 min at 2% B. The runtime per sample including column equilibration, sample loading and gradient separation of peptides was ~60 min. Buffer A was 0.1% formic acid in water and buffer B was 80% acetonitrile, 0.1% formic acid in water.

The mass spectrometer was operated in dia-PASEF mode with a cycle time estimate of 0.96 s. MS1 and MS2 scans were acquired over a mass range from 100 to 1,700 m/z. A method with eight dia-PASEF scans separated into three ion mobility windows per scan covering a 400–1,000 m/z range with 25 D windows and an ion mobility range from 0.64 to 1.37 Vs cm$^2$ was used. Accumulation and ramp time were set to 100 ms, capillary voltage was set to 1,600 V, dry gas was

set to 3 liters/min and dry temperature was set to 200°C. The collision energy was ramped linearly as a function of ion mobility from 59 eV at 1/K0 = 1.6 V s cm$^{-2}$ to 20 eV at 1/K0 = 0.6 V s cm$^{-2}$.

The acquired files were searched using the Spectronaut (Biognosys v19.0) directDIA workflow against a *Mus musculus* database (consisting of 17,085 protein sequences downloaded from Uniprot on 20220222) and 393 commonly observed contaminants. Quantitative fragment ion data (F.Area) were exported from Spectronaut and analyzed using the MSstats R package v.4.13.0. (Choi et al, 2014). Data were normalized using the default normalization option "equalizedMedians," imputed using "AFT model-based imputation" and *P*-values and *q*-values for pairwise comparisons were calculated using the limma package (https://doi.org/10.18129/B9.bioc.limma) (Ritchie et al, 2015). Data are presented in Table S3.

### Statistical analysis

The statistical procedures for the proteomic analyses were conducted following the methods outlined in the respective section. The other statistical analyses were performed using GraphPad Prism 9. Differences between three groups were tested with a one-way analyses of variance (ANOVAs) or two-way ANOVA followed by Tukey's post hoc. For the comparison of two groups, either a two-tailed *t* test, two-way ANOVA with Šídák's multiple comparisons test or a Log-rank (Mantel-Cox) test (the latter only for panel Fig 4B) was used. For the CK levels of WT mice, a ROUT outlier test (Q = 1%) was performed which resulted in the exclusion of 5 of 40 measurements. At least five biological replicates were used for all analyses (except for the proteomic analysis of the Dysf −/− group where one sample had to be excluded because of quality control issues and for the quantification of electron microscopy images). Data are presented as means ± SEM, and number of biological replicates is indicated in the figure legend. A *P*-value (or *q*-value for proteomics) < 0.05 is considered statistically significant.

## Data Availability

Proteomic data have been deposited at the Proteomics Identifications Database (MassIVE, accession number MSV000096818). Published proteomic datasets have also been included in this study (accession number MSV000091401) (Furrer et al, 2023b).

## Supplementary Information

## Acknowledgements

We thank Cinzia Tiberi and Mohamed Chami from the BioEM Lab, Laurent Guerard from the Image Core Facility, the Proteomics Core Facility, and the animal facility caretakers of the Biozentrum for their help as well as Marc Donath and his group from the Department of Biomedicine of the University of Basel for providing equipment and assistance for the insulin

measurement. We would also like to thank Christopher Perry and Corey Osto for their input on the CS activity and Seahorse measurements, respectively. The BLAJ mice were a kind gift of the Jain Foundation. This work has been supported by the Swiss Foundation for Research on Muscle Disease (FSRMM), Jain Foundation, Gottfried and Julia Bangerter-Rhyner-Stiftung, Jubiläumsstiftung von Swiss Life, Siemens Fellowship for Excellence, the Swiss National Science Foundation (CRSII5_209252), and the University of Basel.

## Author Contributions

R Furrer: conceptualization, methodology, investigation, analysis and interpretation, funding acquisition, and writing—original draft.
S Dilbaz: methodology, investigation, and writing—review and editing.
SA Steurer: investigation and writing—review and editing.
G Santos: investigation and writing—review and editing.
B Karrer-Cardel: investigation and writing—review and editing.
D Ritz: methodology, investigation, analysis and interpretation, and writing—review and editing.
M Sinnreich: conceptualization, resources, and writing—review and editing.
C Handschin: conceptualization, methodology, investigation, analysis and interpretation, resources, funding acquisition, supervision, and writing—original draft.

## Conflict of Interest Statement

The authors declare that they have no conflict of interest.

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
