## [Reviewer comments · Life Science Alliance]

Life Science Alliance

Metabolic dysregulation contributes to the development of dysferlinopathy

Regula Furrer, Sedat Dilbaz, Stefan Steurer, Gesa Santos, Bettina Karrer-Cardel, Danilo Ritz, Michael Sinnreich, and Christoph Handschin

DOI: <https://doi.org/10.26508/lsa.202402991>

Corresponding author(s): Christoph Handschin, University of Basel

Review Timeline:

Submission Date:	2024-08-12
Editorial Decision:	2024-09-20
Revision Received:	2025-01-17
Editorial Decision:	2025-02-12
Revision Received:	2025-02-13
Accepted:	2025-02-14

Transaction Report:

September 20, 2024

Re: Life Science Alliance manuscript #LSA-2024-02991-T

Prof. Christoph Handschin
University of Basel
Biozentrum
Spitalstrasse 41
Basel 4056
Switzerland

Dear Dr. Handschin,

Thank you for submitting your manuscript entitled "Metabolic dysregulation contributes to the development of dysferlinopathy" to Life Science Alliance. The manuscript was assessed by expert reviewers, whose comments are appended to this letter. We invite you to submit a revised manuscript addressing the Reviewer comments.

Thank you for this interesting contribution to Life Science Alliance. We are looking forward to receiving your revised manuscript.

Sincerely,

B. MANUSCRIPT ORGANIZATION AND FORMATTING:

Reviewer #1 (Comments to the Authors (Required)):

The manuscript examines glycogen metabolism and the effects of exercise in a mouse model of dysferlin deficiency. The overexpression of PGC1alpha was explored. The use of the PGC1alpha knock out mouse was somewhat justified and the authors state surprising results were seen.

One of the outcomes was that mice given lifelong access to a running wheel were somewhat protected from muscle atrophy possibly through the enhanced glycogen utilization seen with exercise. The translation of this to humans is not relevant as people with dysferlinopathy become confined to a wheelchair within a few years. Of those individuals, there are many who were very active up until the time they could no longer be active. Please explore the literature and establish the translation, or not, of the findings. This is particularly important in "The paper explained" and its relevance.

One of the key findings that the authors should emphasize more than they have currently, is around the need to carefully select the muscles that are being used for investigations into dysferlinopathy. I encourage greater attention to this based on the data presented.

Proteomic data needs validating. This is particularly important as much of the discussion around changes in metabolism rely on these data. Also, the results on page 8 states "This experiment established that exercise-induced glycogen breakdown is not impaired in dysferlin-deficient mice" which is contrary to what would be expected given the apparent protein abundances. Is it that these protein abundances are not as they seem from the proteomic analyses?

Throughout the results, glycogen content needs to be presented in absolute amounts, not fold changes.

Results, page 5. Please provide the validation data to show to what extent PGC1alpha protein is overexpressed in the transgenic animals and the muscle specificity of such. This is important as it is stated that PGC-1a overexpression does not ameliorate disease progression (top page 6), but later that it mitigates muscle fiber membrane damage.

Middle page 7: ..."PGC-1a is not elevated.... "Appendix 2F. change to 'not different'.

Middle page 8, Figure 4: how long did the single bout of exercise last for?

Discussion, page 8: given the varied roles of dysferlin in muscle, I suggest including a broader brush in the opening sentence of the discussion. See On the role of dysferlin in striated muscle: membrane repair, t-tubules and Ca²⁺ handling - Quinn - 2024 - The Journal of Physiology - Wiley Online Library

Top Page 9: "the serendipitous discovery of substantial remodelling of muscle metabolism". There are numerous papers where the link between dysferlin deficiency affects muscle metabolism. This finding is not at all serendipitous. I agree, however, that it goes into greater depth than others have. In saying that, aspects need to be validated (proteins) and presented in absolute values (glycogen) for the relevance of this manuscript as a contribution to the literature. This is a major flaw in the current manuscript.

Top page 10: "high levels of intracellular Ca²⁺ reduce the internalization of GLUT4 (Li et al, 2014)". That citation is in L6 cells and the increase in Ca²⁺ was not physiological. The potential higher Ca²⁺ that would occur with an ionophore would be higher than would occur in vivo.

Please incorporate sex differences into the introduction and discussion and outline limitations in using only male animals. Include details of whether red or white muscles are used when both would be present, e.g. quadriceps.

End of Introduction, page 5. It is stated that data indicate abnormal muscle cell metabolism. This is not novel and relevant literature needs to be incorporated into the introduction. As such, it is not a new aspect of dysferlin biology. It is certainly very

interesting though.

Methods page 15: describe the quantification of Evans Blue dye.
"Quadriceps muscle was removed". From where? Freshly obtained?

Page 16: show that the TATA box-binding protein is a suitable housekeeping gene.
Mass Spectrometry: how much peptide was loaded into Orbitrap? What was the total run time in the MS?

Figure 1

Fig 1D & E: is the y-axis as "% of total fibers"?

Fig 1E: include error bars. Include the muscles used in the figure legend. What is the coefficient of variation for being able to measure the number of Evans blue+ fibers? Please include.

Include mass data for rectus femoris, given it is used in D. Please explain the number of animals at each age used for the individual panels.

Figure 2

The proteomic data presented as specific proteins need to be validated. Include details of how normalized in the figure legend. Define all acronyms used in the figure legend.

Figure 3

Glycogen must be presented in absolute values. Include n values for each test / phenotype / age.

Figure 4

Provide glycogen in absolute values. Use individual data points, as per panel E.

Reviewer #2 (Comments to the Authors (Required)):

The present paper explored the hypothesis that activation of PGC1 α gene could ameliorate the muscle dystrophic phenotype due to dysferlin deficiency. Surprisingly, the authors did not reveal any beneficial action and even a worsening of muscle phenotypes. To dissect the insights of these unexpected findings the authors identified an alteration of glucose homeostasis with an increase Glut expression, glucose blood clearance and increased glycogen content in two dysferlin deficient mouse lines. Finally, authors exercised the dysferlin deficient mice to decrease glycogen overload without improving the CK levels, glycogen content or preventing muscle damage. These observations are interesting and the concept that altered metabolism could play a role in the pathogenetic mechanisms of muscle degeneration when dysferlin is lacking is intriguing. However, the conclusion that glucose metabolism is part of this scenario is not supported by the data. The authors should consider the following points.

Point1. The authors claimed that the accumulation of glycogen is a common feature of glycogen storage disease, which area characterized by myopathic features and weakness. However, this comparison is not appropriate since the accumulation of glycogen is only a consequence and not a cause of the myopathy. Indeed, in GSDII the GAA deficiency cause autophagy-lysosome impairment/dysfunction that is the major trigger of myofiber degeneration with myofiber vacuolization. In GSDIII and IV, the absence of debranching enzymes leads to a bioenergetic defect. To sustain Dysf-GSD similarities authors should check autophagy-lysosome system and ATP levels. Do EM analyses revealed enlarged lysosomes and accumulation of autophagosomes filled with glycogen? Is autophagy flux altered?. Does ATP level drop during exercise? In absence of these data the authors should avoid claiming a pathogenic role of glucose homeostasis in Dysf-/- mice. This is also supported by the discussion in which authors mentioned the negative effect of a low glucose diet in these mice.

Point2. PGC1 α ameliorated the dystrophic phenotype of dystrophin deficient mice because of the fiber type switching and utrophin upregulation (utrophin is higher in slow fibers). However, dysferlin deficiency results in failure of membrane repair with calcium overload. The Ca $^{2+}$ homeostasis was never monitored in these transgenic mice. Is cytosolic Ca $^{2+}$ affected by PGC1 α expression in dysferlin deficient myofibers? It is reasonable that PGC1 α may not ameliorate the dysf-/- phenotype because Ca $^{2+}$ overload persists.

Point3. Mitochondrial morphology (EM analyses) and function should be tested. In case that altered Ca $^{2+}$ homeostasis persists, then mitochondrial function might be affected, and this would explain why having more mitochondria (PGC1 α transgenic) may be detrimental, at least in the context of the quadriceps. More mitochondria, higher risk of death signals coming from Ca $^{2+}$ -induce mitochondrial swelling by PTP opening. Do myofibers contain lipid droplets at EM (a sign of mitochondrial dysfunction)?

Point4. Among the overlap downregulated proteomes, muscle contraction is highly enriched. Which type of proteins are? Are related to EC-coupling? I could not find these data. Authors should consider/discuss their involvement.

Point-by-point response to the reviewers' concerns

Reviewer #1 (Comments to the Authors (Required)):

General remark: We would like to thank the reviewer for his time and suggestions to improve our manuscript. We believe that the new analysis that we performed based on his suggestions further strengthen our manuscript.

General comment: The manuscript examines glycogen metabolism and the effects of exercise in a mouse model of dysferlin deficiency. The overexpression of PGC1alpha was explored. The use of the PGC1alpha knock out mouse was somewhat justified and the authors state surprising results were seen.

Point 1. One of the outcomes was that mice given lifelong access to a running wheel were somewhat protected from muscle atrophy possibly through the enhanced glycogen utilization seen with exercise. The translation of this to humans is not relevant as people with dysferlinopathy become confined to a wheelchair within a few years. Of those individuals, there are many who were very active up until the time they could no longer be active. Please explore the literature and establish the translation, or not, of the findings. This is particularly important in "The paper explained" and its relevance.

REPLY: We agree, and now have more clearly pointed out in the discussion that the exercise intervention is mainly proof-of-concept to assess whether modulating glycogen dynamics and improving mitochondrial function (an analysis that we added based on a point raised by reviewer # 2) has beneficial effects on muscle mass and performance. For potential human translation, these findings could provide pointers towards the underpinnings of health benefits, and thereby help to find novel therapeutic targets. Based on our new mitochondrial data, we now also highlight the importance of exercise mimetics not merely increasing mitochondrial volume density, but rather enhancing mitochondrial function in a coordinated and balanced manner, which seems critical for cellular health in dysferlin-deficient mice. Of note, some form of exercise or neuromuscular functional rehabilitation therapy programs could (and should) still be considered in the treatment of LGMDs (D'Este *et al*, 2025), obviously choosing protocols that minimize fiber damage. Wheel chair-dependence per se should not rule out a training program as long as some muscles are still exhibiting a certain degree of functionality.

Point 2. One of the key findings that the authors should emphasize more than they have currently, is around the need to carefully select the muscles that are being used for investigations into dysferlinopathy. I encourage greater attention to this based on the data presented.

REPLY: We would like to thank the reviewer for pointing this out. Indeed, this is a very important finding and critical to include in future studies (and interpret the results reported in prior manuscripts). We now highlighted these findings in the results and the discussion section.

Point 3. Proteomic data needs validating. This is particularly important as much of the discussion around changes in metabolism rely on these data. Also, the results on page 8 states "This experiment established that exercise-induced glycogen breakdown is not impaired in dysferlin-deficient mice" which is contrary to what would be expected given the apparent protein abundances. Is it that these protein abundances are not as they seem from the proteomic analyses?

REPLY: As suggested by the reviewer, we now have validated our proteomics data in a different mouse model. We believe that showing the same metabolic alterations regarding the downregulation of glycolytic proteins and glycogen phosphorylase concomitant with an elevation of GLUT4 and G6PD in BLAJ mice (which also have elevated muscle glycogen) strengthens our point in respect to altered glucose and glycogen metabolism (Fig. 3L & Figs. S7E & F).

In regard to the statement that “experiment established that exercise-induced glycogen breakdown is not impaired in dysferlin-deficient mice” even when this is not apparent in protein abundance, we have the following explanation and additional data:

- We performed phospho-proteomics to assess whether at rest the phosphorylation state of GSK3 β , GYS1 and PYGM could shed light on the activity of these enzymes. However, the analysis revealed a mixed picture. When phospho-data are normalized to total protein (Fig S6D), phosphorylation of inhibiting GSK3 β site Ser9 is higher in the absence of dysferlin. In line with this result, phosphorylation of the inhibiting site Ser645 of GYS1 is lower, concomitant with a reduced phosphorylation of various sites of PYGM (except the activating site Ser15), suggesting an enhanced synthesis and decreased breakdown of glycogen. However, when not normalizing the data to total protein (Fig. S6E), Ser9 phosphorylation of GSK3 β is lower while phosphorylation of Ser645 of GYS1 and various sites of PYGM are still lower. Of note, there are various phosphorylation sites that are altered for which the function is unclear. Collectively, these data nevertheless point towards an elevated synthesis and lower breakdown of glycogen. Ultimately however, regardless of the individual impact of protein levels and phosphorylation events, a net effect is on glycogen content is clearly found, e.g. as shown in Figure 3G-I.
- The proteomics shows the resting state of the muscle and therefore, it is difficult to draw conclusions regarding the exercise effect. Thus, we also performed proteomics and phospho-proteomics of the muscle post-exercise. Interestingly, inhibition and activation of enzymes involved in glycogen synthesis and breakdown are distinct between WT and Dysf $-/-$ muscles. First, protein levels of PYGM are only increased in Dysf $-/-$ muscles after an acute bout of exercise to exhaustion (Fig. S8A). Second, the phosphorylation of GYS1 and PYGM are affected differently upon exercise. For example, many phospho-sites of PGYM exhibit elevated phosphorylation after exercise in Dysf $-/-$ mice, while this is not the case in WT mice, in which some phospho-site even display lower phosphorylation (Figs. S8B & C). Despite the distinct activation of the enzymes, the net effect on glycogen breakdown appears to be similar.

Point 4. Throughout the results, glycogen content needs to be presented in absolute amounts, not fold changes.

REPLY: As suggested, in the revised version of the manuscript, glycogen content is presented in absolute amounts showing the individual values.

Point 5. Results, page 5. Please provide the validation data to show to what extent PGC1alpha protein is overexpressed in the transgenic animals and the muscle specificity of such. This is important as it is stated that PGC-1a overexpression does not ameliorate disease progression (top page 6), but later that it mitigates muscle fiber membrane damage.

REPLY: The currently available PGC-1 α antibodies have issues detecting PGC-1 α in a sensitive and specific manner (see Figure below). Therefore, we measured PGC-1 α RNA

expression in WT, *Dysf*^{-/-}, *Dysf*^{-/-} PGC-1 α and PGC-1 α muscles. PGC-1 α expression is ~7-fold higher in PGC-1 α mice compared to WT and ~5-fold higher in *Dysf*^{-/-} PGC-1 α (Fig. S1A). The PGC-1 α mouse line expressed the transgene under the control of the muscle creatine kinase (MCK) promoter and has been well characterized (Lin *et al*, 2002). The authors of the original paper demonstrate that MCK-driven gene expression is highest in skeletal muscles, and much lower in cardiac muscles, while being very low or non-detectable in other tissues (Johnson *et al*, 1989). Indeed, we previously also assessed PGC-1 α expression in heart and showed that the expression is double that of WT mice (Whitehead *et al*, 2018), but still substantially lower compared to skeletal muscles.

Figure. Western blots of skeletal muscle of wild type (WT) animals, mice overexpressing PGC-1 α specifically in muscle (mTg PGC-1 α) and mice lacking PGC-1 α (global knockouts; PGC-1 α gKO) using two different PGC-1 α antibodies from OriGene (TA343708) and Millipore (ST1202). With both antibodies, we also obtain a band in the PGC-1 α gKO sample (negative control), indicating that there is aspecific binding.

Point 6. Middle page 7: ... "PGC-1 α is not elevated.... "Appendix 2F. change to 'not different'.

REPLY: We changed this to “the expression is not different in dysferlin-deficient mice compared to WT (Fig. S1A)” as suggested.

Point 7. Middle page 8, Figure 4: how long did the single bout of exercise last for?

REPLY: The exercise bout was performed until the mice were exhausted and therefore the time was slightly different for each mouse. We plotted the time that mice ran until they reached exhaustion (Fig. 4B), and display the individual values in order to be able to see the exact length of the exercise bout for each mouse.

Point 8. Discussion, page 8: given the varied roles of dysferlin in muscle, I suggest including a broader brush in the opening sentence of the discussion. See On the role of dysferlin in striated muscle: membrane repair, t-tubules and Ca²⁺ handling - Quinn - 2024 - The Journal of Physiology - Wiley Online Library

REPLY: We have expanded the discussion and added the reference as suggested.

“Dysferlin is a central mediator of vesicle-vesicle and vesicle-membrane fusion, and thus instrumental for damage mitigation, in particular in a mechanically challenged tissue such as skeletal muscle (Han & Campbell, 2007). However, in the past two decades, dysferlin emerged as a multifaceted protein that is not only involved in membrane repair but also plays a role in the maintenance of t-tubule integrity, the regulation of excitation-contraction coupling and Ca²⁺ handling (Quinn et al., 2024), Therefore, ...”

Point 9. Top Page 9: "the serendipitous discovery of substantial remodelling of muscle metabolism". There are numerous papers where the link between dysferlin deficiency affects muscle metabolism. This finding is not at all serendipitous. I agree, however, that it goes into greater depth than others have. In saying that, aspects need to be validated (proteins) and presented in absolute values (glycogen) for the relevance of this manuscript as a contribution to the literature. This is a major flaw in the current manuscript.

REPLY: We have rephrased this sentence and included more papers describing alterations in metabolism. As mentioned in point 3 and 4, we have validated our proteomics data in a different dysferlinopathy mouse model (BLAJ) and now also present glycogen content as absolute values.

Point 10. Top page 10: "high levels of intracellular Ca²⁺ reduce the internalization of GLUT4 (Li et al, 2014)". That citation is in L6 cells and the increase in Ca²⁺ was not physiological. The potential higher Ca²⁺ that would occur with an ionophore would be higher than would occur in vivo.

REPLY: We have now added to this section in the discussion that intracellular Ca²⁺ levels induced by ionomycin may be substantially higher than those observed after injury, which should be considered when interpreting the data.

Point 11. Please incorporate sex differences into the introduction and discussion and outline limitations in using only male animals. Include details of whether red or white muscles are used when both would be present, e.g. quadriceps.

REPLY: We have added the information about the specific use of male mice to the introduction and discussion, have emphasized that our findings apply to male mice and that it is unclear whether all results also apply to female mice. Furthermore, we have added a statement about the specificity of the disease manifestation in terms of red and white muscles: “Hence, there is no disease preference towards a certain fiber type (fast- vs slow-twitch muscle fibers) but rather towards location (proximal vs distal).” With regard to the quadriceps muscle, we have specified in the methods section that we used muscle homogenate for the analysis.

Point 12. End of Introduction, page 5. It is stated that data indicate abnormal muscle cell metabolism. This is not novel and relevant literature needs to be incorporated into the introduction. As such, it is not a new aspect of dysferlin biology. It is certainly very interesting though.

REPLY: We now have added various papers on dysregulated metabolism to the discussion section.

Point 13 - Methods

1. *Methods page 15: describe the quantification of Evans Blue dye.*
2. *"Quadriceps muscle was removed". From where? Freshly obtained?*
3. *Page 16: show that the TATA box-binding protein is a suitable housekeeping gene.*

4. *Mass Spectrometry: how much peptide was loaded into Orbitrap? What was the total run time in the MS?*

REPLY: We have described the quantification of Evans blue in more detail and added more information about the removal of the quadriceps muscle, the selection of TBP as a housekeeping gene, and the the mass spectrometry analysis.

1. Quantification of Evans blue dye: To assess muscle damage, the Evans blue positive muscle fibers of the rectus femoris of the quadriceps muscle were manually counted.
2. Removal: Immediately after confirming death induced by CO₂ overdose, blood was collected from the vena cava in tubes containing lithium heparin (Sarstedt) and plasma prepared. Subsequently, the following hind limb muscles were freshly removed, weighted, snap-frozen in liquid nitrogen and stored at -80°C for further analysis: psoas, gluteus, quadriceps, TA, EDL and soleus muscle. For mass spectrometry, immunoblotting, qPCR, glucose uptake, glycogen content, Ca²⁺ concentration, citrate synthase activity and mitochondrial respiration (using Seahorse) measurements, the quadriceps muscle was pulverized prior to further processing. Hence, the muscle tissue homogenate contained a proportional mixture of all abundant fiber types. The quadriceps muscle of the other limb was additionally embedded in 7% tragacanth (Sigma-Aldrich), snap-frozen in liquid nitrogen-cooled isopentane (-150°C) and stored at -80°C.
3. TBP as housekeeper: *Tbp* was found to be a suitable housekeeping genes, with mean ct values of 25.569 for WT (range 25.266-25.996), 25.621 for *Dysf* *-/-* (range 25.303-26.168), 25.621 for PGC-1α mTG (range 25.305-25.824) and 26.599 for *Dysf* *-/-* PGC-1α (range 26.229-27.066). Only *Dysf* *-/-* PGC-1α was different to the other groups, but even when we tested *S18* as a housekeeper, the ct value of the *Dysf* *-/-* PGC-1α group was also 1 cycle later compared to that of the other three groups that were almost identical (ct of ~12.2 in *Dysf* *-/-* PGC-1α vs ~11.2 in other samples), suggesting that the cDNA concentration is lower in the samples of the *Dysf* *-/-* PGC-1α group.
4. Mass spectrometry: For the proteomic analysis of WT, *Dysf* *-/-*, PGC-1α mTG, *Dysf* *-/-* PGC-1α, WT exercise and *Dysf* *-/-* exercise, 250 ng peptide were injected on column per sample and the AGC target of the orbitrap was set to 1e6 with a maximum injection time of 50 ms. The runtime per sample including column equilibration, sample loading and gradient separation of peptides was approx. 140 min. for the Phosphoproteomic analysis, 300 ug protein were digested and resulting peptides were subjected to phosphopeptide enrichment. 15% of the phosphoenriched peptide preparation were injected on column per sample. The AGC target of the orbitrap was set to 1e6 with a maximum injection time of 50 ms for MS1 scans and to 5e4 and 54 ms for MS2 scans. The runtime per sample including column equilibration, sample loading and gradient separation of peptides was approx. 130 min. And for the validation experiment with the BLAJ mice, 50 ng peptide were injected on column per sample. The runtime per sample including column equilibration, sample loading and gradient separation of peptides was approx. 60 min.

Point 14 - Figures

Figure 1

Fig 1D & E: is the y-axis as "% of total fibers"?

Fig 1E: include error bars. Include the muscles used in the figure legend. What is the coefficient of variation for being able to measure the number of Evans blue+ fibers? Please include. Include mass data for rectus femoris, given it is used in D. Please explain the number of animals at each age used for the individual panels.

REPLY: The y-axis of Fig. 1D shows the total number of fibers counted in the rectus femoris muscle. Fig. 1E (of the previous version of the manuscript) did show a relative representation of the data. However, we decided to remove Fig. 1E, since this was only a different representation of Fig. 1D and did not add any additional value. We have only assessed the weight of the complete quadriceps muscle (all “four heads”), which is significantly affected by the pathology, and not only or specifically of the rectus femoris.

Figure 2

The proteomic data presented as specific proteins need to be validated. Include details of how normalized in the figure legend. Define all acronyms used in the figure legend.

REPLY: We have validated our proteomics finding in a different mouse model for dysferlinopathy (Fig. 3L & Figs. S7E & F). Additionally, we now have added more details about the normalization and defined all abbreviations.

Figure 3

Glycogen must be presented in absolute values. Include n values for each test / phenotype / age.

REPLY: We now display all individual data points and present the absolute values. For line figures (in which we were not able to show individual points), we provide detailed information about the n for each genotype and age in the figure legend.

Figure 4

Provide glycogen in absolute values. Use individual data points, as per panel E.

REPLY: We have changed the figure accordingly.

References

- D'Este G, Spagna M, Federico S, Cacciante L, Cieslik B, Kiper P, Barresi R (2025) Limb-girdle muscular dystrophies: A scoping review and overview of currently available rehabilitation strategies. *Muscle Nerve* 71: 138-146
- Johnson JE, Wold BJ, Hauschka SD (1989) Muscle creatine kinase sequence elements regulating skeletal and cardiac muscle expression in transgenic mice. *Molecular and cellular biology* 9: 3393-3399
- Lin J, Wu H, Tarr PT, Zhang CY, Wu Z, Boss O, Michael LF, Puigserver P, Isotani E, Olson EN *et al* (2002) Transcriptional co-activator PGC-1 alpha drives the formation of slow-twitch muscle fibres. *Nature* 418: 797-801
- Whitehead N, Gill JF, Brink M, Handschin C (2018) Moderate Modulation of Cardiac PGC-1alpha Expression Partially Affects Age-Associated Transcriptional Remodeling of the Heart. *Frontiers in physiology* 9: 242

Reviewer #2 (Comments to the Authors (Required)):

General remark: We really appreciate the time the reviewer took to read our manuscript and provide constructive feedback. We performed a series of new experiments to address the involvement of Ca²⁺ overload mitochondrial dysfunction death signaling to shed light on this disease aspect and believe that this substantially improved the manuscript.

General comment: *The present paper explored the hypothesis that activation of PGC1a gene could ameliorate the muscle dystrophic phenotype due to dysferlin deficiency. Surprisingly, the authors did not reveal any beneficial action and even a worsening of muscle phenotypes. To dissect the insights of these unexpected findings the authors identified an alteration of glucose homeostasis with an increase Glut expression, glucose blood clearance and increased glycogen content in two dysferlin deficient mouse lines. Finally, authors exercised the dysferlin deficient mice to decrease glycogen overload without improving the CK levels, glycogen content or preventing muscle damage. These observations are interesting and the concept that altered metabolism could play a role in the pathogenetic mechanisms of muscle degeneration when dysferlin is lacking is intriguing. However, the conclusion that glucose metabolism is part of this scenario is not supported by the data. The authors should consider the following points.*

Point 1. *The authors claimed that the accumulation of glycogen is a common feature of glycogen storage disease, which area characterized by myopathic features and weakness. However, this comparison is not appropriate since the accumulation of glycogen is only a consequence and not a cause of the myopathy. Indeed, in GSDII the GAA deficiency cause autophagy-lysosome impairment/dysfunction that is the major trigger of myofiber degeneration with myofiber vacuolization. In GSDIII and IV, the absence of debranching enzymes leads to a bioenergetic defect. To sustain Dysf-GSD similarities authors should check autophagy-lysosome system and ATP levels. Do EM analyses revealed enlarged lysosomes and accumulation of autophagosomes filled with glycogen? Is autophagy flux altered?. Does ATP level drop during exercise? In absence of these data the authors should avoid claiming a pathogenic role of glucose homeostasis in Dysfer^{-/-} mice. This is also supported by the discussion in which authors mentioned the negative effect of a low glucose diet in these mice.*

REPLY: We have now carefully analyzed our EM images, however did not observe a clear accumulation of enlarged lysosomes or glycophaosomes, and therefore have no indication that autophagy is impaired in dysferlin-deficient muscles. With regard to the question whether ATP drops during exercise, we demonstrate that WT and Dysf^{-/-} mice run for the same duration, and have a similar maximal oxygen uptake and respiratory exchange ratio (RER) throughout the running experiment (Figs. S9A & B). Furthermore, lactate levels post-exercise were comparable to WT mice (Fig. S9C), suggesting that energy provisioning did not differ between these genotypes. Furthermore, to substantiate these findings, we have also performed phospho-proteomics of the muscles collected immediately post-exercise. We now have added this data to the manuscript (Figs. S9D-H) and show that the metabolic stress (or altered metabolic demand) in response to exercise is very similar in WT and Dysf^{-/-} mice, for example reflected by a comparable phosphorylation of AMPK and the downstream substrates ACC1 and ACC2.

Although we still mention that elevated muscle glycogen can induce muscle abnormalities and present the data of the diet intervention (because we believe that it is important to let others know that this diet is not effective in dysferlinopathies), we now refrain from a direct comparison of dysferlinopathy and GSD. We now have also added various experiments as proposed below and could now show that the general metabolic dysregulation, including mitochondrial abnormalities, could contribute to the pathology, e.g. as indicated by the activation of the cGAS-STING pathway (Fig. 2H).

Point 2. PGC1 α ameliorated the dystrophic phenotype of dystrophin deficient mice because of the fiber type switching and utrophin upregulation (utrophin is higher in slow fibers). However, dysferlin deficiency results in failure of membrane repair with calcium overload. The Ca²⁺ homeostasis was never monitored in these transgenic mice. Is cytosolic Ca²⁺ affected by PGC1 α expression in dysferlin deficient myofibers? It is reasonable that PGC1 α may not ameliorate the dysf^{-/-} phenotype because Ca²⁺ overload persists.

REPLY: This is a very interesting point raised by the reviewer. Unfortunately, we could not directly assess Ca²⁺ kinetics in the different genotypes. Instead, we have measured total Ca²⁺ concentration and determined the abundance of proteins involved in Ca²⁺ homeostasis. Therefore, we also have added proteomic data of Dysf^{-/-} PGC-1 α to the manuscript. Together, these data reveal that total Ca²⁺ concentration is higher in muscles overexpressing PGC-1 α independent of dysferlin (Fig. S5A). Then, in regard to the proteins involved in Ca²⁺ handling, Dysf^{-/-}, Dysf^{-/-} PGC-1 α and PGC-1 α all showed the same patterns (Fig. 2G & Fig. S5B). For example, there was a substantially higher abundance of MCU, which points towards a mitochondrial phenotype in response to dysregulated Ca²⁺ homeostasis (see point 3).

Point 3. Mitochondrial morphology (EM analyses) and function should be tested. In case that altered Ca²⁺ homeostasis persists, then mitochondrial function might be affected, and this would explain why having more mitochondria (PGC1 α transgenic) may be detrimental, at least in the context of the quadriceps. More mitochondria, higher risk of death signals coming from Ca²⁺-induce mitochondrial swelling by PTP opening. Do myofibers contain lipid droplets at EM (a sign of mitochondrial dysfunction)?

REPLY: We thank the reviewer for this excellent suggestion. Indeed, addressing this point revealed very intriguing results that strengthened our manuscript and support the hypothesis of substantial metabolic remodeling observed in the absence of dysferlin. First, we have quantified mitochondrial number and size in the EM images and observed that while mitochondrial volume density is similar in WT and Dysf^{-/-} mice, Dysf^{-/-} have less, but larger mitochondria (Figs. 2C-E), supporting the hypothesis of mitochondrial swelling by PTP opening. Despite the same volume density, the abundance of mitochondrial proteins is reduced and function tends to be lower (Fig. 2F & Figs. S3C-E), suggesting that the mitochondria are dysfunctional. We did not observe any changes in lipid droplets. In mice overexpressing PGC-1 α , there was no difference in volume density, number and size of the mitochondria between Dysf^{-/-} PGC-1 α and PGC-1 α mice (Figs. S4A-C). However, there was a substantial reduction in mitochondrial proteins and mitochondrial function as indicated by citrate synthase activity and cellular respiration (complex II and IV), both of which were drastically decreased, reaching similar levels to WT animals, despite higher muscle PGC-1 α (Figs. S4D-G). This suggest that while mitochondrial density is unchanged in Dysf^{-/-} PGC-1 α , their function is impaired, which could be detrimental for cellular health and provide another explanation why the disease is exacerbated in these mice. Furthermore, the results imply that at least for some functions, PGC-1 α relies on adequate dysferlin levels. Such an interaction

has not been reported so far. To further unravel the hypothesis about Ca²⁺ overload, mitochondrial swelling and potential death signaling, we have also analyzed the activation level of the cGAS-STING pathway. We could demonstrate that this pathway is considerably elevated in muscles lacking dysferlin (Fig. 2H). Therefore, we can now show that in addition to the dysregulation of glucose and glycogen metabolism, the lack of dysferlin promote mitochondrial abnormalities leading to elevated death signaling.

Point 4. Among the overlap downregulated proteomes, muscle contraction is highly enriched. Which type of proteins are? Are related to EC-coupling? I could not find these data. Authors should consider/discuss their involvement.

REPLY: The term “muscle contraction” of Fig. 2B mainly contains contractile proteins related to fast-twitch muscles that show a change in expression between the genotypes. A potential shift in fiber type (or at least fiber-type typical properties) is also reflected in the lower abundance of SERCA1, CASQ1 and PVALB and the concomitant increase in CASQ2. We now have added these data to the manuscript (Fig. S5C).

February 12, 2025

RE: Life Science Alliance Manuscript #LSA-2024-02991-TR

Prof. Christoph Handschin
University of Basel
Biozentrum
Spitalstrasse 41
Basel 4056
Switzerland

Dear Dr. Handschin,

Thank you for submitting your revised manuscript entitled "Metabolic dysregulation contributes to the development of dysferlinopathy". We would be happy to publish your paper in Life Science Alliance pending final revisions necessary to meet our formatting guidelines.

- please be sure that the authorship listing and order is correct
- please add the X and Bluesky handles of your host institute/organization as well as your own or/and one of the authors in our system
- please note that the titles in the system and manuscript file must match
- please add a Conflict of Interest statement to your main manuscript text
- please add your main, supplementary figure, and movie legends to the main manuscript text after the references section and remove the SI file with this info
- you may want to consider uploading Figure 5 as a Graphical Abstract rather than as a figure, but this is up to you

A. FINAL FILES:

B. MANUSCRIPT ORGANIZATION AND FORMATTING:

Sincerely,

Reviewer #2 (Comments to the Authors (Required)):

The paper is improved. The authors addressed my concerns. I have no further requests.

February 14, 2025

RE: Life Science Alliance Manuscript #LSA-2024-02991-TRR

Prof. Christoph Handschin
University of Basel
Biozentrum
Spitalstrasse 41
Basel 4056
Switzerland

Dear Dr. Handschin,

Thank you for submitting your Research Article entitled "Metabolic dysregulation contributes to the development of dysferlinopathy". It is a pleasure to let you know that your manuscript is now accepted for publication in Life Science Alliance. Congratulations on this interesting work.

DISTRIBUTION OF MATERIALS:

Again, congratulations on a very nice paper. I hope you found the review process to be constructive and are pleased with how the manuscript was handled editorially. We look forward to future exciting submissions from your lab.

Sincerely,
